# Transfer Learning-Assisted Evolutionary Dynamic Optimisation for Dynamic Human-Robot Collaborative Disassembly Line Balancing

Liang Jin [1], Xiao Zhang [2], Yilin Fang [2,*] and Duc Truong Pham [3]

1   School of Mechanical and Electronic Engineering, Wuhan University of Technology, Wuhan 430070, China
2   School of Information Engineering, Wuhan University of Technology, Wuhan 430070, China
3   School of Engineering, University of Birmingham, Birmingham B15 2TT, UK
*   Correspondence: fangspirit@whut.edu.cn

**Abstract:** In a human-robot collaborative disassembly line, multiple people and robots collaboratively perform disassembly operations at each workstation. Due to dynamic factors, such as end-of-life product quality and human capabilities, the line balancing problem for the human-robot collaborative disassembly line is a dynamic optimisation problem. Therefore, this paper investigates this problem in detail and commits to finding the evolutionary dynamic optimisation. First, a task-based dynamic disassembly process model is proposed. The model can characterise all feasible task sequences of disassembly operations and the dynamic characteristics of tasks affected by uncertain product quality and human capabilities. Second, a multiobjective optimisation model and a feature-based transfer learning-assisted evolutionary dynamic optimisation algorithm for the dynamic human-robot collaborative disassembly line balancing problem are developed. Third, the proposed algorithm uses the balanced distribution adaptation method to transfer the knowledge of the optimal solutions between related problems in time series to track and respond to changes in the dynamic disassembly environment. Then, it obtains the optimal solution sets in a time-varying environment in time. Finally, based on a set of problem instances generated in this study, the proposed algorithm and several competitors are compared and analysed in terms of performance indicators, such as the mean inverted generational distance and the mean hypervolume, verifying the effectiveness of the proposed algorithm on dynamic human-robot collaborative disassembly line balancing.

**Keywords:** dynamic disassembly line balancing; human-robot collaboration; disassembly process model; evolutionary dynamic optimisation; transfer learning





## 1. Introduction

End-of-life (EOL) products have the dual attributes of environmental pollution and resource regeneration. Improper disposal will harm the ecological environment, while proper disposal can bring substantial economic benefits. The remanufacturing lifecycle of EOL products includes disassembly, cleaning, inspection, repair, reassembly, and testing. Disassembly is one of the most critical stages of the lifecycle, and it is also the basis for the realisation of remanufacturing goals. The disassembly system gradually separates the usable parts from EOL products following the predetermined objectives and following various constraints, which is usually regarded as the reverse process of the assembly system. Many researchers are currently conducting disassembly-related research based on the ideas from many preliminary works of the past 50 years in the assembly field. The disassembly system contains the same essential elements as the assembly system. However, due to the uncontrolled quality of EOL products entering the industrial environment, some parts of a product may be missing, damaged, or worn. Moreover, the supply of products and the demand for parts may be time-varying. These factors will significantly increase the uncertainty in the disassembly system compared to the assembly system [1].

Since the 1990s, robot-based disassembly systems have received extensive attention from the academic community, and researchers have carried out much work on automated disassembly cells. However, these works are mainly oriented to the structured environment, and they cannot deal with various uncertainties, which significantly limits their industrial applications. With the development of information technologies, such as ubiquitous perception, industrial robots, and artificial intelligence, autonomous disassembly systems that can effectively tackle environmental uncertainties have gradually attracted attention and made progress [2]. In disassembly systems, a disassembly line composed of multiple workstations connected in sequence through a transmission module is an operation mode that is suitable for remanufacturing enterprises to implement product disassembly. Compared to the disassembly cell, it can guarantee higher productivity. Building the autonomous disassembly system represented by the robotic disassembly line is a long-term goal of disassembly research. Robots have significant advantages over humans in strength, speed, repeatability, and stamina. However, due to the low flexibility of robots and their limited cognitive capabilities, some complex and sophisticated tasks still require humans to complete them [3]. Moreover, the dynamic structure of the robot is complex [4], and no robot can handle all possible tasks. Human-robot collaboration can effectively utilise the complementary advantages between robots and humans, and it will be an important mode for product disassembly. Therefore, constructing a human-robot collaborative disassembly line (HRDL) that can deal with various environmental uncertainties should be a feasible direction in the present and future disassembly fields.

The disassembly line balancing problem (DLBP) is a crucial planning problem in the disassembly line. It considers how to allocate a set of disassembly operations to the operators in the workstations to ensure that the constraints are not violated, while also optimising various predetermined performance measures [5]. In the HRDL, multiple humans and robots collaboratively perform operations at a workstation. The human uncertainty caused by this mode and the inherent EOL product quality uncertainty result in DLBP being a dynamic optimisation problem. Additionally, the uncertainty of product quality leads to possible differences between batches. We define a batch as a disassembly environment, with multiple different batches corresponding to multiple disassembly environments, constituting a dynamic disassembly environment in terms of time sequence. To simplify the problem, we assume that the product state remains constant in a single disassembly environment. Therefore, this paper studies the dynamic human-robot collaborative disassembly line balancing (D-HRDLB) problem so the constructed HRDL can track these uncertain factors and calculate the optimal solutions in the new environment.

The literature lacks a study on the D-HRDLB problem, and the existing theories in the disassembly field are not well applicable to the problem. In the DLBP literature, the task precedence diagram (TPD) [6] and the transformed AND/OR graph (TAOG) [7] are two widely used disassembly process models for modelling disassembly operations and the precedence relationship between operations. Both models assume that the product information extracted from the original CAD model is known and fixed. This assumption is derived from and only applies to the assembly scenario. The disassembly scenario does not hold because the product quality is uncertain. A disassembly operation is the division of a subassembly into two subassemblies, representing a feasible cut-set on the connection diagram of the product. A disassembly operation includes the disconnection tasks of all edges in its cut-set and various auxiliary tasks. Generally, one disassembly operation corresponds to multiple feasible task sequences. Existing disassembly process models, such as the TPD and the TAOG, take the disassembly operation as the basic element, which makes it difficult to characterise all feasible task sequences for disassembly operations. Moreover, these disassembly process models are static or quasi-static models, and they do not consider the time-varying nature of attributes, such as the feasibility of the task sequence that is affected by the uncertain product quality. To bridge this gap, this study proposes a dynamic disassembly process model based on tasks. This proposed model can

characterise all feasible task sequences of disassembly operations and the time-varying characteristics of tasks affected by uncertain product quality and human capabilities.

In the uncertain DLBP literature, most of the existing methods dealing with uncertainty propose some nondeterministic models and algorithms based on statistical theories, such as stochastic numbers [8], fuzzy numbers [9], and interval numbers [10], for some uncertain parameters (e.g., the disassembly operation processing times). These methods need to presuppose appropriate statistical functions or know some statistical information, which makes them difficult to implement in industrial applications. Moreover, they cannot track multisource uncertain factors, such as product quality and human capabilities in real time. Therefore, it is difficult to respond to environmental changes. To fill this gap, this study proposes a solution method for the D-HRDLB problem based on evolutionary dynamic optimisation [11]. Evolutionary optimisation [12] is a metaheuristic method based on population evolution that is suitable for solving multiobjective combinatorial optimisation problems with irregular Pareto fronts, such as the DLBP. Evolutionary dynamic optimisation is based on evolutionary optimisation by further adding fast convergence strategies, such as generating a good initial population, so the optimal solution set changing with the dynamic environment can be obtained in time. This dynamic optimisation method can track and respond to various environmental uncertainties over time. At present, there is no research on dynamic DLBPs using this method.

From the challenges described above, this paper examines the D-HRDLB problem in detail. Humans and robots can perform one or more disassembly operations collaboratively at any workstation of a disassembly line. To characterise the human-robot collaborative disassembly mode and dynamic factors, such as uncertain product quality, a task-based dynamic disassembly process model is proposed. Based on the dynamic disassembly process model, a multiobjective optimisation model of the D-HRDLB problem is developed. In the D-HRDLB problem, there is a correlation between the problems in different environments. Therefore, this study adopts the data distribution adaptation method [13] in feature-based transfer learning to generate a good initial population in a new environment. Furthermore, a transfer learning-assisted evolutionary dynamic optimisation algorithm is proposed to solve the D-HRDLB problem. To verify the effectiveness of the proposed algorithm, a new set of test instances for the D-HRDLB problem is generated, and the proposed algorithm is compared with several competitors in terms of the mean inverted generational distance (MIGD) and the mean hypervolume (MHV). Compared with the existing studies on DLBP, the main differences and contributions of this study are fourfold.

(1) The D-HRDLB problem is introduced in this paper. This problem fully considers the characteristics of human-robot collaboration, and it can track and respond to the dynamic changes in the disassembly environment in real time. Compared with the traditional DLBPs, this problem is more applicable to the actual production environment in the remanufacturing enterprise.

(2) A task-based dynamic disassembly process model is proposed. This model takes disassembly tasks as the basic elements instead of the traditional disassembly operations, and it can characterise the time-varying characteristics of the task, such as the task feasibility affected by the uncertain product quality.

(3) A mathematical model for the D-HRDLB problem with three objectives is developed to optimise small-scale problem instances.

(4) A feature-based transfer-assisted evolutionary dynamic optimisation algorithm is developed to obtain a dynamic Pareto-optimal solution set for the multiobjective optimisation of the D-HRDLB problem. The algorithm can transfer the knowledge of optimal solution sets between similar problems in different environments, thereby tracking and responding to environmental changes and obtaining the dynamic optimal solution set in a time-varying environment.

The importance of this research is mainly reflected in the following two aspects. First, the human-robot cooperative disassembly line balance problem proposed by us is a dynamic multiobjective optimisation problem considering the uncertainty of product

quality, which is more in line with the actual disassembly environment. Second, the optimisation algorithms used to solve the DLBP problem usually target a deterministic disassembly environment. The algorithm proposed in this study can effectively deal with the changing disassembly environment, and it has more practical significance.

The remainder of this paper is structured as follows. Section 2 discusses the related work from two aspects: the disassembly process model and the DLBP under uncertainty. Subsequently, Section 3 presents and defines the related concepts of the task-based dynamic disassembly process model. The concept of the D-HRDLB problem is explained, and a three-objective optimisation model for this problem is developed in Section 4. Section 5 delivers insights into the implementation details of the proposed algorithm, including the basic algorithm framework, initial population generation, solution encoding and decoding, and solution variation. Afterwards, experimental studies are provided in Section 6. Finally, Section 7 concludes the study and highlights possible future research avenues.

## 2. Literature Review

### 2.1. Disassembly Process Model

To solve the D-HRDLB problem, a task-based dynamic disassembly process model that can track the time-varying environment is needed. The EOL product modelling methods in DLBP literature are derived from the assembly field, including the connection diagram and its enhanced versions, such as the datum flow chain [14] and the relational model for assemblies [15]. The connection diagram shows the relations between parts, but it does not describe the space-blocking relations. However, the blocking relation is a necessary input to generate or detect feasible disassembly operations. Therefore, product structure models, such as the directional blocking graph, the nondirectional blocking graph [16], and the interference matrix [17], have been proposed one after another. Reasoning on the structure of an EOL product and the blocking relations between parts, all the detachable subassemblies or parts, as well as all the corresponding feasible disassembly operations and precedence relationships between operations, are obtained. Disassembly process models, including TPDs, TAOGs, state diagrams [18], hierarchical trees [19], and Petri nets [20], are generally used to describe this information. In addition, some complete product information modelling methods have been proposed to define product structural and behavioural schema that can reflect all production information [21]. Furthermore, no disassembly process model considers product quality, human capabilities, and other uncertain factors.

### 2.2. Disassembly Line Balancing under Uncertainty

Although there is a high level of environmental uncertainty in the DLBP, few studies in the DLBP literature consider the uncertain environment. These studies follow the idea of statistical modelling; establish nondeterministic models for specific uncertain parameters, such as the operation processing time, demand, revenue, and workload at the workstations; and use stochastic numbers [22–24], fuzzy numbers [25–27], interval numbers [10], and other nondeterministic approaches to address uncertainty in DLBP models. He et al. [28] studied a bi-objective stochastic disassembly line balancing problem to minimise the line design cost and the cycle time, using only the knowledge of the mean, standard deviation, and upper bound of stochastic task processing times. Based on the characteristics of the DLBP, a method based on fuzzy set theory is developed to evaluate the performance scores and determine the ranking of disassembly tasks [27]. Some of these studies establish relationships between uncertain product quality and one of the above uncertain parameters. For instance, Colledani, and Battaïa [29] studied DLBP considering different product quality categories. It was assumed that the operation processing time under different categories was a random variable obeying a normal distribution. Bentaha, Voisin, and Marangé [30] studied the relation between the condition of a subassembly and the disassembly revenue, and they used a stochastic approach to model revenue to solve the nondeterministic disassembly sequence planning. Bentaha, Moalla, and Ouzrout [31]

also established a link between product condition and revenue, creating a disassembly line that can handle product quality uncertainty. Zhu et al. [32] established a connection between operation processing time and product condition based on a probability-weighted approach. These studies need to presuppose appropriate statistical functions or know some statistical information. Their time-varying abilities are limited to updating statistics, such as probability. Moreover, they are based on disassembly operations rather than tasks, making it challenging to estimate uncertain parameters in real time. In addition, Altekin and Akkan [33] and Mete et al. [34] studied the DLBP in the case of disassembly operation failure and supply change, respectively, but they did not consider the uncertainty of product quality. Although some existing research considers the uncertainty of product quality, they usually transform the uncertain problem into a deterministic one, which cannot effectively deal with the dynamic environment in actual disassembly. The D-HRDLB problem proposed in this paper combines multiple disassembly environments into a set of dynamic environments in time order, which is closer to the real-life disassembly scenario.

Research on the human-robot collaborative disassembly line has just started in recent years. Xu et al. [35] studied the disassembly information model of human-robot collaborative disassembly considering the safe strategy. Likewise, Luca et al. [36] improved the safety of the operators working in a collaborative workspace. Natalia et al. [37] pointed out that human-robot collaborative solutions allow operators to coexist and interact safely with robots of various payloads. Additionally, Liu et al. [38] and Li et al. [39] studied the influence of the distance between humans and robots and human fatigue on the operation processing time, belonging to the scope of deterministic disassembly planning. Most human-robot collaborative assembly studies adopt timeline-based planning approaches to cope with human uncertainty. However, even if based on timelines, most planners cannot manage uncontrollable events and human factors in real time [3]. Therefore, this study develops a solution method for the D-HRDLB problem based on evolutionary dynamic optimisation to track environmental changes and obtain the dynamic optimal solution set in real time.

Furthermore, DLBPs with multiple uncertainty factors often have multiple conflicting optimisation objectives. Therefore, multiobjective optimisation algorithms are usually used to solve DLBPs, and they have a wide range of applications. Ouadfel et al. [40] presented a multiobjective weighted, multiview clustering method based on a gradient-based optimiser. Additionally, Sharifi et al. [41] proposed the multiobjective moth swarm algorithm for the first time to solve various multiobjective problems. To handle the multiobjective optimisation problems of truss-bar design, Premkumar et al. [42] introduced a new metaheuristic multiobjective optimisation algorithm. Fox et al. [43] described an efficient connectivity-based method for multiobjective optimisation applicable to the design of marine protected area networks. Finally, Dhiman et al. [44] extended the Seagull Optimisation Algorithm to deal with multiobjective problems.

## 3. Task-Based Dynamic Disassembly Process Model

In this section, a task-based dynamic disassembly process model is developed to depict all the potential disassembly sequences of completely separating a product into its components and consider the uncertainty of the disassembly process. Before discussing the model, some related terminologies need to be defined. The basic elements of a product include the components that make up the product and the connections between them. These elements are generally represented using a connection diagram. A disassembly operation is the detachment of a component or a subassembly, represented by a cut-set in the corresponding connection diagram. A disassembly process is a sequence of disassembly operations through which a product is separated into its components. Restricted by topological constraints, geometric constraints, and technical constraints of the product, these operations need to be performed in a specific order. These orderly operations in the disassembly process are also known as disassembly sequences.

In general, numerous possible disassembly sequences can be applied to disassemble a product completely. The transformed AND/OR graph (TAOG) [7] is a frequently used disassembly process model representing all these possible disassembly sequences and precedence relationships among all operations in these sequences. In the TAOG representation, nodes are divided into artificial and normal nodes. Artificial nodes represent subassemblies and components, while normal nodes correspond to disassembly operations. The precedence relationships between artificial nodes and normal nodes are represented by the arcs connecting the nodes. Two types of hyperarcs are emphasised to differentiate between AND relationships and OR relationships. An AND-type hyperarc is adjacent to one normal node and two artificial nodes. A disassembly operation can split a product or subassembly into two subassemblies. An OR-type hyperarc is adjacent to one artificial node and adjacent to one or more normal nodes, which indicates that one or more follow-up disassembly operations can be selected, and only one of these operations should be processed.

Using a TAOG as the input to the DLB problem implies that the selection of disassembly sequences and the assignment of the selected disassembly sequence are considered simultaneously. However, the TAOG representation ignores the fact that each disassembly operation is complex, requiring multiple humans or robots to work collaboratively in multiple steps. A disassembly operation must be subdivided into multiple elementary tasks to facilitate execution. These tasks involve not only some disassembly tasks, such as disconnecting all connections in the corresponding cut-set, but also some additional tasks, such as component state detection, fixture establishment, robot moving, tool changing, and product transferring. A proper disassembly process model should be able to represent tasks rather than disassembly operations alone, and DLB problems should consider the assignment of tasks rather than the assignment of disassembly operations alone. In addition, used products may contain rusty, deformed, or missing components, together with the variable ability and uncertain behaviour of humans, which determine that the disassembly process is dynamic and uncertain. The TAOG representation cannot model these characteristics of the disassembly process. For the reasons given above, we developed another disassembly process model called a task-based dynamic TAOG (TD-TAOG). It is an extended version of a TAOG containing all possible disassembly operations, and all possible tasks can also consider the dynamics and the uncertainty of the disassembly process.

A TD-TAOG contains complete information about the dynamic disassembly process of a product. TD-TAOG is divided into the operation layer and the task layer. The operation layer corresponds to the TAOG of the product. For each disassembly operation (normal node) on the operation layer, there is a task graph corresponding to it on the task layer. The task graph contains all possible task sequences for completing the operation. Note that a task sequence consists of multiple elementary tasks that jointly complete an operation. Unlike a disassembly operation, which is always feasible, whether a task in a task sequence is feasible depends on the current state of the component or subassembly to be disassembled. For example, if the component to be disassembled is detected as damaged, tasks used in the normal state of the component are not feasible, and they cannot be chosen to complete the corresponding operation. The state of a component needs to be identified by sensor feedback. In addition, some task processing times in TD-TAOG are time-varying due to the influence of uncertain human abilities and behaviours. Clearly, the introduction of the task graph for each operation gives the possibility of processing various dynamic operations intuitively and practically, considering the uncertainty of the disassembly process.

TD-TAOG contains three types of nodes, artificial nodes, normal nodes, and unit nodes, with two types of hyperarcs: AND-type hyperarcs and OR-type hyperarcs. Among them, the definitions of artificial nodes, normal nodes, AND-type hyperarcs, and OR-type hyperarcs are the same as the corresponding definitions in the TAOG. Unit nodes in the task layer represent elementary tasks, including disassembly and supplementary tasks. To distinguish these two types of hyperarcs, we use a slight curve as the mark of the OR-type hyperarcs. A flashlight [45] is taken as an example of a product to better illustrate

TD-TAOG. The flashlight is constructed with seven components and their connections, as shown in Figure 1. Figure 1 contains the cross-section view and the connection diagram of the flashlight. The TD-TAOG of the flashlight in the $\ell$th environment is given in Figure 2. In Figure 2, subassemblies, disassembly operations, and tasks are modelled by artificial nodes labelled $A_a$s, normal nodes labelled $B_b$s, and unit nodes labelled $U_u$s. The number sequence next to each artificial node represents the related subassembly. For instance, artificial node $A_6$ represents the subassembly '5/7', which consists of component five (main housing), component six (spring), and component seven (battery). For simplicity, in Figure 2, subassemblies with only one component are not shown.

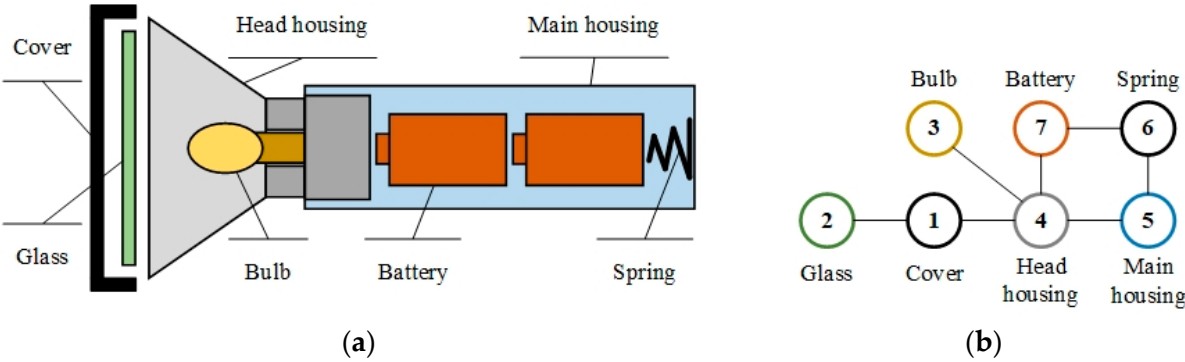

(**a**)                                                                                 (**b**)

**Figure 1.** An example of a product: (**a**) a flashlight [45] and (**b**) the connection diagram of the flashlight.

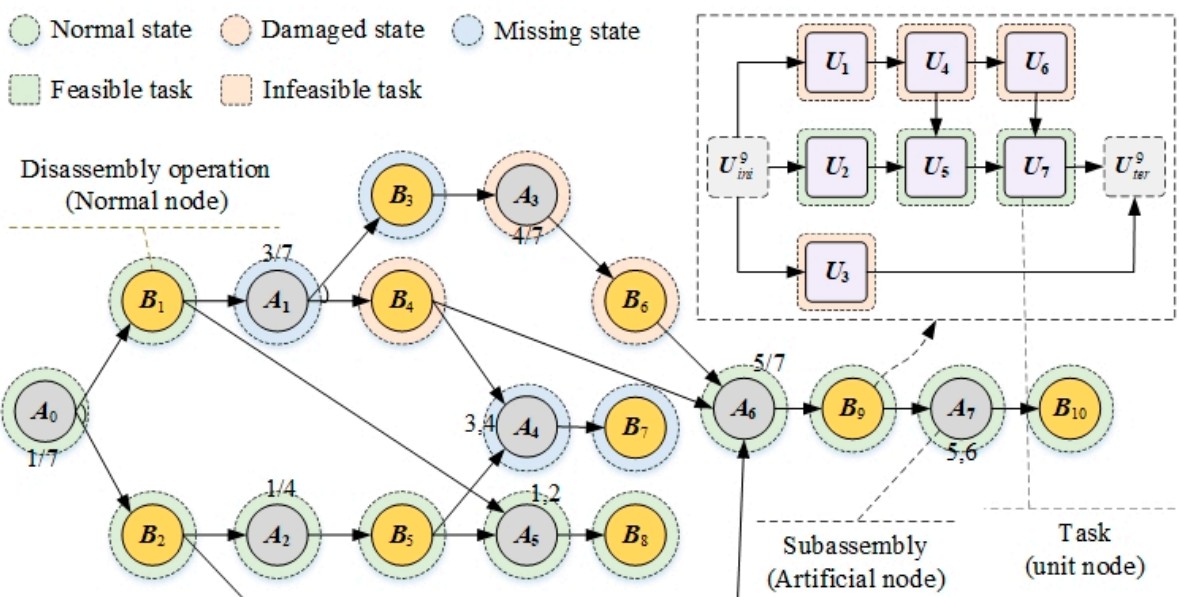

**Figure 2.** The TD-TAOG of the flashlight in the $\ell$th environment.

Furthermore, only the task graph of operation $B_9$ is shown in Figure 2. The operation layer has four AND-type hyperarcs and two OR-type hyperarcs. In addition, Figure 2 provides numerous possible disassembly sequences, such as the disassembly sequence $[B_1, B_8, B_4, B_7, B_9,$ and $B_{10}]$. Based on feedback from the sensors, the state of an artificial node or a normal node can be determined in real time. In the example shown in Figure 2, since component three (bulb) is detected as being lost at the starting time instant of the $\ell$th environment, nodes $A_1$, $A_4$, $B_3$, and $B_7$ are determined to be in the missing state. Component four (head housing) is detected as damaged; therefore, nodes $A_3$, $B_4$, and $B_6$ are determined to be in the damaged state. In the task graph of operation $B_9$, $U_{ini}^9$ and $U_{ter}^9$ are the initial and terminal nodes, respectively. These two dummy nodes (dummy tasks) are introduced to obtain feasible task sequences. Each directed arc in the task graph represents

an ordered task pair. We need to find a path consisting of feasible unit nodes (tasks) from $U_{ini}{}^9$ and $U_{ter}{}^9$ as a feasible task sequence to complete $B_9$. In Figure 2, since $B_9$ is in the normal state, unit nodes for the normal state, such as $U_2$, $U_5$, and $U_7$, are all feasible.

In contrast, unit nodes such as $U_1$, $U_4$, $U_6$, and $U_3$ are infeasible for damaged or missing states. Therefore, these unit nodes cannot be selected to complete $B_9$. The feasibility of each disassembly task is determined by the state of the disassembly operation to which it belongs. Those feasible tasks in the damaged state can only be selected when the disassembly operation they belong to is in the damaged state. Note that there may be multiple feasible task sequences in each state of a disassembly operation. The processing time of some tasks in the task graph is time-varying.

## 4. Dynamic Human-Robot Collaborative Disassembly Line Balancing

This section addresses a serial-paced HRDL and one of its dynamic line-balancing problems. The HRDL is formed by multiple human-robot collaborative workstations connected in a series by a conveyor system. Multiple types of used products in the same batch can enter the HRDL for complete disassembly at the same time. The disassembly operations are assigned to each workstation in sequence to disassemble these products completely, according to their precedence relationships. The state of each component in the same product is allowed to change in different batches. We assume that they can be identified by sensor feedback before the disassembly of the product begins.

It is specified that multiple robots and humans can work at the same workstation. They share the same workspace and resources. Considering the workspace and resource constraints, there is an upper limit to the number of robots and humans that can be accommodated at each workstation. For each disassembly operation assigned to a workstation, at least one feasible task sequence in its corresponding task graph can be used to complete it at any time. Generally, humans have high flexibility, while robots are superior in strength and accuracy and more suitable for performing unsafe tasks. In the HRDL, a task can only be assigned to a robot or a human that meets its requirement.

The D-HRDLB problem is raised to balance the HRDL to work efficiently in different environments. This problem is assumed to be stationary for each environment. It contains four subproblems for each environment: (a) how to find the optimal disassembly sequence for each product; (b) how to find the optimal task sequence for each selected disassembly operation; (c) how to assign the selected tasks to the appropriate robots or humans; and (d) how to allocate the robots and the humans that have been assigned tasks to an ordered sequence of workstations. These four subproblems need to be optimised simultaneously under a series of constraints, such as precedence and sequence constraints. In this study, three optimisation objectives are considered: minimising the cycle time, minimising the total number of robots and humans in use, and minimising the number of humans in use. In addition, the number of workstations is fixed and predetermined.

Figure 3 shows an example of a D-HRDLB problem in the $\ell$th environment. In Figure 3, an HRDL consists of three human-robot collaborative workstations. A used flashlight is completely disassembled on this line. At the starting time instant of the $\ell$th environment, the multimodel sensing system detects that component three (bulb) of the flashlight is lost; thus, the HRDL changes from the $(\ell - 1)$th environment to the $\ell$th environment. Since the selected task sequence for the disassembly operation—seven ($B_7$) in the $(\ell - 1)$th environment—is no longer feasible, for the $\ell$th environment, the planning system needs to redetermine the feasible task sequences for $B_7$ and recalculate the Pareto-optimal solutions. One of the Pareto-optimal solutions in the $\ell$th environment is shown in the lower part of Figure 3. The D-HRDLB problem in changing environments can be formally defined by a dynamic multiobjective optimisation model. In the formulation, the feasibility of a task and the set of operators used to perform a task are taken as dynamic factors leading to changing environments.

The indices, sets and parameters, and decision variables for modelling the D-HRDLB problem are given in Table 1, Table 2, and Table 3, respectively. To simplify the model expression, the TD-TAOGs corresponding to multiple types of products are merged into a general TD-TAOG by default.

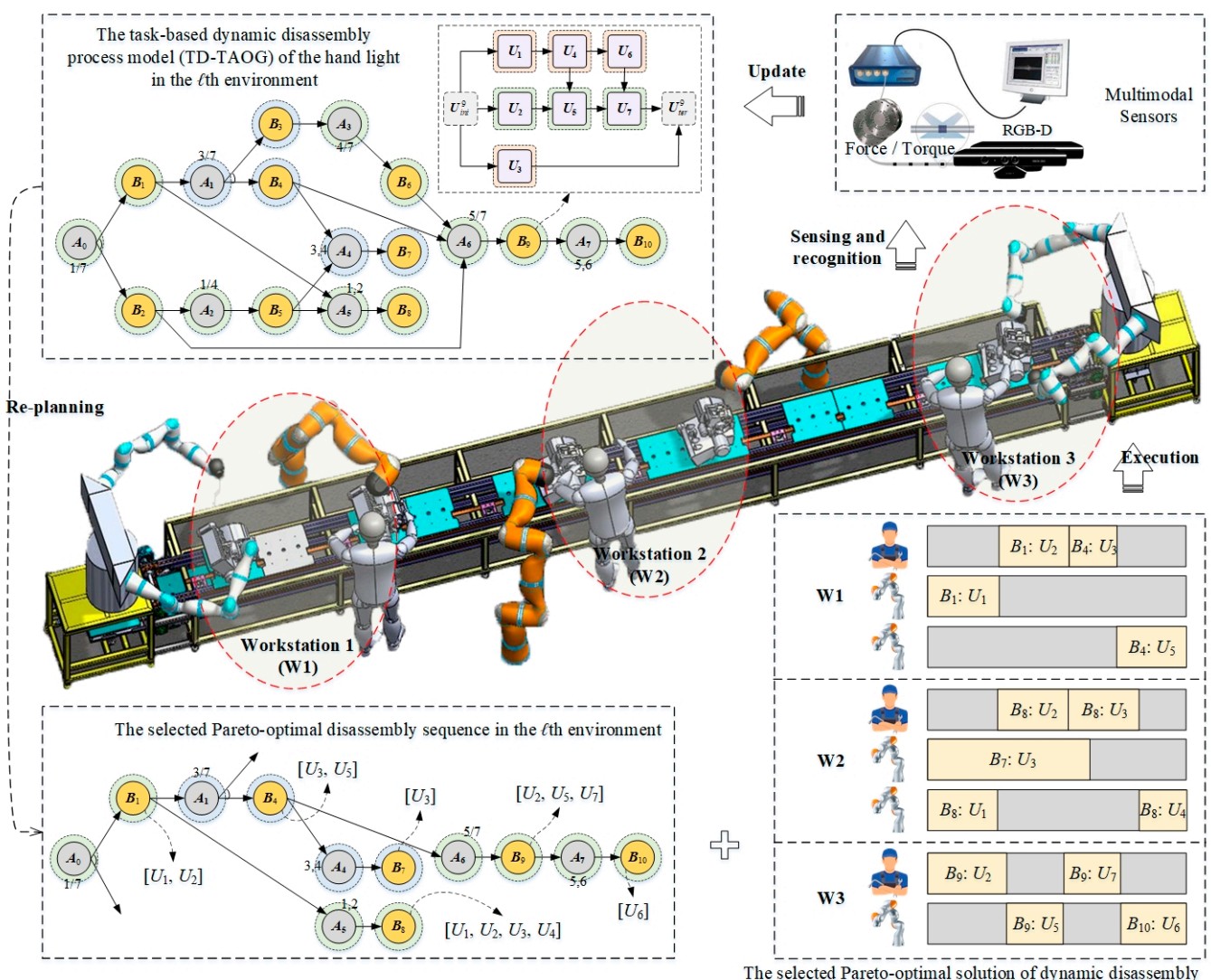

**Figure 3.** Example of a D-HRDLB problem in the $\ell$th environment.

$$\min CT(\ell) \tag{1}$$

$$\min NO(\ell) = \sum_{W_w \in \mathbb{W}} \sum_{O_o \in \mathbb{R} \cup \mathbb{H}} L_{wo} \tag{2}$$

$$\min NH(\ell) = \sum_{W_w \in \mathbb{W}} \sum_{O_o \in \mathbb{H}} L_{wo} \tag{3}$$

Formula (1)–(3) give the three objective functions for the model in any environment, including minimising the cycle time, the total number of robots and humans being used, and the number of humans being used. Notably, there is no need to consider workload balancing between humans and robots in the human-robot collaborative disassembly environment. Therefore, Formula (3) is an objective function.

**Table 1.** The indices used in the model.

| Notation | Definition | Notation | Definition |
|----------|-----------|----------|-----------|
| $a$ | Index of artificial nodes | $b, b'$ | Index of normal nodes |
| $u, u', v, v', \sigma, \varsigma$ | Index of unit nodes | $w, w'$ | Index of human-robot collaborative workstations |
| $o, o'$ | Index of operators | $t, t'$ | Index of time |
| $\ell$ | Index of environments | | |

**Table 2.** The sets and parameters used in the model.

| Notation | Definition | Notation | Definition |
|----------|-----------|----------|-----------|
| $\mathbb{W}$ | Set of workstations | $\mathbb{R}$ | Set of robots |
| $\mathbb{H}$ | Set of humans | $\mathbb{T}$ | Set of time |
| $\mathbb{I}(\ell)$ | Set of all selected unit nodes in the $\ell$th environment | $TO$ | Maximum number of operators at each workstation |
| $Ro(\mathbb{A}(\ell))$ | Set of artificial root nodes | $S(A_a)$ | Set of immediate successors of $A_a$ |
| $P(A_a)$ | Set of immediate predecessors of $A_a$ | $FS(U_u, \ell)$ | Set of feasible immediate successors of $U_u$ in the $\ell$th environment |
| $FP(U_u, \ell)$ | Set of feasible immediate predecessors of $U_u$ in the $\ell$th environment | $F(\mathbb{R} \cup \mathbb{H}, U_u, \ell)$ | Set of feasible operators for $U_u$ in the $\ell$th environment |
| $IP(U_u, \mathbb{I}(\ell))$ | Set of immediate predecessors of $U_u$ in $\mathbb{I}(\ell)$ | $IP_a(U_u, \mathbb{I}(\ell))$ | Set of all predecessors of $U_u$ in $\mathbb{I}(\ell)$ |
| $IS_a(U_u, \mathbb{I}(\ell))$ | Set of all successors of $U_u$ in $\mathbb{I}(\ell)$ | $W_w$ | The $w$th workstation |
| $O_o$ | The $o$th operator | $\psi$ | A big positive number |
| $TC_{uo}(\ell)$ | The time required for the $o$th operator to process $U_u$ in the $\ell$th environment | $|\mathbb{I}(\ell)|$ | The number of unit nodes in $\mathbb{I}(\ell)$ |
| $comt_u(\ell)$ | Completion time of $U_u$ in the $\ell$th environment | | |

**Table 3.** The decision variables used in the model.

| Notation | Definition | Notation | Definition |
|----------|-----------|----------|-----------|
| $Z_b$ | 1, if $B_b$ is selected to perform; 0, otherwise | $G_{bv}$ | 1, if $U_v \in \mathbb{U}^b(\ell) \cup \left\{ U_{ini}^b, U_{ter}^b \right\}$ is selected to complete $B_b$; 0, otherwise |
| $X_{uwot}$ | 1, if $U_u$ is assigned to the $o$th operator (e.g., a feasible robot or human for $U_u$) in the $w$th workstation and completes its processing at time $t$; 0, otherwise | $L_{wo}$ | 1, if the $o$th operator is used in the $w$th workstation; 0, otherwise |
| $Y_{uu'}$ | 1, if $U_u$ is executed earlier than $U_{u'}$ by the same operator; 0, otherwise | | |

$$comt_u(\ell) \leq CT(\ell), \ \forall U_u \in \mathbb{I}(\ell) \tag{4}$$

Formula (4) defines the cycle time constraint.

$$\sum_{B_b \in S(A_a)} Z_b = 1, \ \forall A_a \in Ro(\mathbb{A}(\ell)) \tag{5}$$

$$\sum_{B_b \in S(A_a)} Z_b = \sum_{B_{b'} \in P(A_a)} Z_{b'}, \ \forall A_a \in \mathbb{A}(\ell) - Ro(\mathbb{A}(\ell)) \tag{6}$$

$$Z_b \odot G_{bv} = 1, \ \forall B_b \in \mathbb{B}(\ell) \ and \ \forall U_v \in \{U_{ini}^b, U_{ter}^b\} \tag{7}$$

$$Z_b \odot \left( \sum_{U_v \in FS(U_{ini}^b, \ell)} G_{bv} \right) = Z_b \odot \left( \sum_{U_{v'} \in FP(U_{ter}^b, \ell)} G_{bv'} \right) = 1, \ \forall B_b \in \mathbb{B}(\ell) \tag{8}$$

$$G_{bu} \cdot \left(1 - \sum_{U_\sigma \in FS(U_u, \ell)} G_{b\sigma}\right) = G_{bu} \cdot \left(1 - \sum_{U_\varsigma \in FP(U_u, \ell)} G_{b\varsigma}\right) = 0, \ \forall B_b \in \mathbb{B}(\ell) \ and \ \forall U_u \in \mathbb{U}^b(\ell) \tag{9}$$

$$\sum_{W_w \in \mathbb{W}} \sum_{O_o \in F(\mathbb{R} \cup \mathbb{H}, U_u, \ell)} \sum_{t \in \mathbb{T}} X_{uwot} = G_{bu}, \ \forall B_b \in \mathbb{B}(\ell) \ and \ \forall U_u \in \mathbb{U}^b(\ell) \tag{10}$$

Formulae (5) and (6) ensure that for each product, only one disassembly sequence can be selected from the operation layer of its TD-TAOG for the complete disassembly of the product. Formulae (7)–(9) guarantee that for each operation in the disassembly sequence, only one feasible task sequence can be chosen from the task layer of the TD-TAOG to complete the operation. Formula (10) ensures that a selected task can only be assigned to one of the feasible operators at one of the workstations and finished at a precise time.

$$Z_b \cdot G_{bu} \cdot G_{bu'} \cdot \left( \sum_{W_w \in \mathbb{W}} \sum_{O_o \in F(\mathbb{R} \cup \mathbb{H}, U_u, \ell)} \sum_{t \in \mathbb{T}} w \cdot X_{uwot} - \sum_{W_{w'} \in \mathbb{W}} \sum_{O_{o'} \in F(\mathbb{R} \cup \mathbb{H}, U_{u'}, \ell)} \sum_{t' \in \mathbb{T}} w' \cdot X_{u'w'o't'} \right) = 0,$$
$$\forall B_b \in \mathbb{B}(\ell) \ and \ \forall U_u, U_{u'} \in \mathbb{U}^b(\ell) \tag{11}$$

Formula (11) checks the integrity constraints of the selected operations. In other words, all selected tasks for completing the same operation must be assigned to the same workstation.

$$Z_b \cdot Z_{b'} \cdot G_{bu} \cdot G_{b'u'} \cdot \left( \sum_{W_w \in \mathbb{W}} \sum_{O_o \in F(\mathbb{R} \cup \mathbb{H}, U_u, \ell)} \sum_{t \in \mathbb{T}} w \cdot X_{uwot} - \sum_{W_{w'} \in \mathbb{W}} \sum_{O_{o'} \in F(\mathbb{R} \cup \mathbb{H}, U_{u'}, \ell)} \sum_{t' \in \mathbb{T}} w' \cdot X_{u'w'o't'} \right) \leq 0,$$
$$\forall A_a \in \mathbb{A}(\ell) - Ro(\mathbb{A}(\ell)) \ and \ \forall B_b \in P(A_a) \ and \ \forall B_{b'} \in S(A_a) \ and \ \forall U_u \in \mathbb{U}^b(\ell) \ and \ \forall U_{u'} \in \mathbb{U}^{b'}(\ell) \tag{12}$$

Formula (12) ensures that the assignment of the selected tasks cannot violate the precedence constraints between tasks.

$$comt_u(\ell) - comt_{u'}(\ell)$$
$$+ \psi \cdot (1 - \sum_{O_o \in F(\mathbb{R} \cup \mathbb{H}, U_u, \ell)} \sum_{t \in \mathbb{T}} X_{uwot}) + \psi \cdot (1 - \sum_{O_o \in F(\mathbb{R} \cup \mathbb{H}, U_{u'}, \ell)} \sum_{t \in \mathbb{T}} X_{u'wot}) \geq \sum_{O_o \in F(\mathbb{R} \cup \mathbb{H}, U_u, \ell)} \sum_{t \in \mathbb{T}} X_{uwot} \cdot TC_{uo}(\ell),$$
$$\forall U_u \in \mathbb{I}(\ell) \ and \ U_{u'} \in IP(U_u, \mathbb{I}(\ell)) \ and \ \forall W_w \in \mathbb{W} \tag{13}$$

$$comt_u(\ell) - comt_{u'}(\ell) + \psi \cdot (1 - X_{uwot}) + \psi \cdot (1 - X_{u'w't'}) + \psi \cdot (1 - Y_{u'u}) \geq X_{uwot} \cdot TC_{uo}(\ell),$$
$$\forall U_u \in \mathbb{I}(\ell) \ and \ \forall U_{u'} \in \mathbb{I}(\ell) \backslash \{IP_a(U_u, \mathbb{I}(\ell)) \cup IS_a(U_u, \mathbb{I}(\ell))\} \ and \ \forall W_w \in \mathbb{W} \ and \ \forall t \in \mathbb{T}$$
$$and \ \forall O_o \in F(\mathbb{R} \cup \mathbb{H}, U_u, \ell) \ and \ \forall O_{o'} \in F(\mathbb{R} \cup \mathbb{H}, U_{u'}, \ell) \tag{14}$$

Formulas (13) and (14) control the sequence-dependent completion time of each selected task. For any pair of tasks with an immediate precedence constraint on the same workstation, Formula (13) becomes active. For a pair of selected tasks that have no precedence constraint, but are assigned to the same operator at the same workstation, Formula (14) becomes active.

$$\mathbb{I}(\ell) = \{U_u | B_b \in \mathbb{B}(\ell) \ and \ U_u \in \mathbb{U}^b(\ell) \ and \ Z_b = G_{bu} = 1\} \tag{15}$$

$$comt_u(\ell) = \sum_{W_w \in \mathbb{W}} \sum_{O_o \in F(\mathbb{R} \cup \mathbb{H}, U_u, \ell)} \sum_{t \in \mathbb{T}} t \cdot X_{uwot}, \ \forall U_u \in \mathbb{I}(\ell) \tag{16}$$

$$comt_u(\ell) - \sum_{W_w \in \mathbb{W}} \sum_{O_o \in F(\mathbb{R} \cup \mathbb{H}, U_u, \ell)} \sum_{t \in \mathbb{T}} X_{uwot} \cdot TC_{uo}(\ell) \geq 0, \ \forall U_u \in \mathbb{I}(\ell) \tag{17}$$

$$TC_{uo}(\ell) > 0, \ \forall U_u \in \mathbb{U}(\ell) \ and \ \forall O_o \in F(\mathbb{R} \cup \mathbb{H}, U_u, \ell) \tag{18}$$

$$\sum_{U_u \in \mathbb{I}(\ell)} X_{uwot} \leq |\mathbb{I}(\ell)| \cdot L_{wo}, \ \forall W_w \in \mathbb{W} \ and \ \forall O_o \in \mathbb{R} \cup \mathbb{H} \ and \ \forall t \in \mathbb{T} \tag{19}$$

$$1 \leq \sum_{O_o \in \mathbb{R} \cup \mathbb{H}} L_{wo} \leq TO, \ \forall W_w \in \mathbb{W} \tag{20}$$

$$\sum_{W_w \in \mathbb{W}} L_{wo} \leq 1, \ \forall O_o \in \mathbb{R} \cup \mathbb{H} \tag{21}$$

$$FS(U_{ini}^b, \ell), FP(U_{ter}^b, \ell), FS(U_u, \ell), FP(U_u, \ell), F(\mathbb{R} \cup \mathbb{H}, U_u, \ell) \neq \varnothing,$$
$$\forall B_b \in \mathbb{B}(\ell) \ and \ \forall U_u \in \mathbb{U}^b(\ell) \cap \mathbb{I}(\ell) \tag{22}$$

$$Z_b, G_{bv}, X_{uwot}, L_{wo}, Y_{uu'}, RE_{uor} \in \{0, 1\} \tag{23}$$

Formula (15) defines the set of all selected tasks, and Formula (16) defines the completion time of each selected task. Formulae (17) and (18) define the value ranges of task completion time and processing time, respectively. Formula (19) ensures that the values of Xuwot and Lwo remain consistent. Formula (20) states that the number of operators permitted to allocate to a workstation is between 1 and TO. Formula (21) specifies that each selected operator can only be allocated one workstation. Formula (22) ensures that at least one feasible solution exists in the model. Finally, Formula (23) defines the value range of each decision variable.

**5. Transfer Learning-Assisted Dynamic Evolutionary Algorithm**

Jiang et al. [46] proposed a transfer learning-based dynamic multiobjective optimisation algorithm. This approach exploits the transfer learning technique as a tool to generate an effective initial population pool by reusing past experience to speed up the evolutionary process. In the D-HRDLB problem, the feasible region and the POS in the decision space and the POF in the objective space vary with changing environments. To address these issues, we propose a feature-based transfer learning-assisted dynamic evolutionary algorithm called B-DMOEA. The target of B-DMOEA is to generate an excellent initial population in a new environment by reusing historical POSs, thereby accelerating the evolutionary process to track the movement of the dynamic POS. If the POSs and the POFs of the D-HRDLB problem under different environments are correlated, the good solutions in the historical POSs will bring significant benefits to tracking the POS in the new environment. Figure 4 shows the logic flow chart for B-DMOEA. Algorithm 1 shows the procedure of B-DMOEA, in which an initial population is constructed randomly. Then, the POS and the POF of the D-HRDLB problem under the initial environment ($\ell = 0$) are determined by using a multiobjective evolutionary algorithm (MOEA). If a new environment is detected, the environment index $\ell$ is updated as $\ell = \ell + 1$. Then, an initial population generation method (B-IPG) based on transfer learning with balanced distribution adaptation (BDA) [13] is utilised to generate an excellent initial population that can be used by the MOEA to search for the POF in the new environment. In the following sections, each component of B-DMOEA will be presented in detail. Note that any population-based MOEA can be embedded in B-DMOEA.

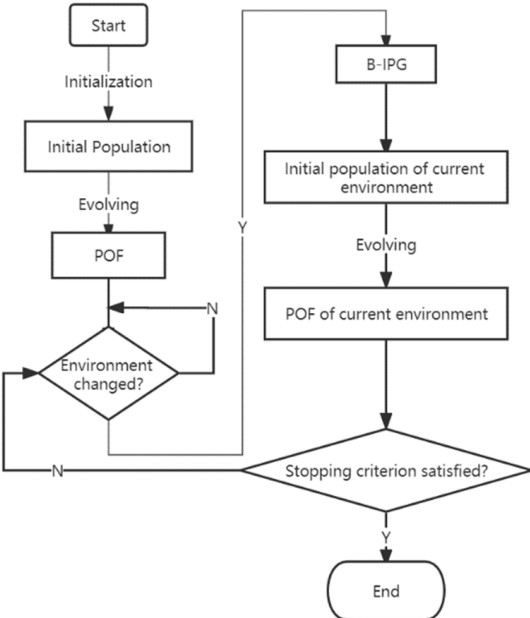

**Figure 4.** Logic flow chart for B-DMOEA.

---

**Algorithm 1** B-DMOEA

---

**Input**: D-HRDLB problem in the $\ell$th environment ($FC_\ell$, $\ell = 0, 1 \ldots$ ); a MOEA; kernel function $\kappa$.
**Output**: The POFs of the D-HRDLB problem.
1: Initialise the environment index $\ell = 0$;
2: Initialise randomly a population *init_pop* with population size $N$;
3: Output $POF_0 \leftarrow$ **MOEA**(*init_pop*, $FC_0$);
4: **If** (*the environment has changed*) **Then**
5:    $\ell = \ell + 1$;
6:    *init_pop* $\leftarrow$ **B-IPG**($POF_0, \ldots, POF_{\ell-1}, FC_0, \ldots, FC_\ell, \kappa$);
7:    Output $POF_\ell \leftarrow$ **MOEA**(*init_pop*, $FC_\ell$);
8: **End If**
9: **If** (the stopping criterion is satisfied) **Then** Stop; **Else** Go to Line 4;

---

*5.1. Transfer Learning-Assisted Initial Population Generation*

We take the objective space in the current environment as the target domain, and then we select the objective space in the historical environments with the slightest discrepancy in the marginal and conditional distributions from the target domain as the source domain. Our idea is to use BDA to find an optimal feature transformation so the marginal and conditional distribution discrepancies between domains are as small as possible in the latent space obtained by the transformation. Furthermore, each nondominated solution in the POF of the source domain is re-represented in the latent space. Therefore, the corresponding solution found in the target domain can be as close to it as possible after being transferred to the latent space. These solutions are used to construct the initial population.

The diversity of samples in the source and target domains is important to reduce the probability of negative transfer and improve the quality of the initial population. Therefore, unlike traditional random sampling, we perform sample presearch based on MOEAs to increase the diversity of the samples. The sample distribution can represent the actual distribution of domains as much as possible [47]. BDA is adopted to learn the feature transformation once the domain sampling is completed, inspired by Jiang et al. [46]. In the D-HRDLB problem, the source and target domains have different marginal and conditional distributions, but the distributions between the domains are correlated. In addition, the marginal and conditional distributions are not equally important for knowledge transfer. Our goal is to find an optimal feature transformation that makes the difference in the interdomain weighted distribution based on the marginal and conditional distributions as small as possible. A nonparametric distance estimation method called maximum mean discrepancy (MMD) differentiates distributions in the reproducing kernel Hilbert space (RKHS). The weighted distribution discrepancy based on MMD is defined as follows.

$$
\begin{aligned}
\mathrm{MMD}(D_s, D_t) \doteq\ & (1 - \mu) \cdot \left\| \frac{1}{m} \sum_{x_i \in D_s} \phi(x_i) - \frac{1}{n} \sum_{x_j \in D_t} \phi(x_j) \right\|_{\mathcal{H}}^2 \\
& + \mu \cdot \sum_{c=0}^{1} \left\| \frac{1}{m_c} \sum_{x_i \in D_s^c} \phi(x_i) - \frac{1}{n_c} \sum_{x_j \in D_t^c} \phi(x_j) \right\|_{\mathcal{H}}^2
\end{aligned}
\tag{24}
$$

where $\phi(.) : \mathcal{D} \rightarrow \mathcal{H}$ represents the mapping from the objective space ($\mathcal{D}$) to the RKHS ($\mathcal{H}$); $m$ and $n$ represent the number of samples in the sample sets of the source domain ($D_s = \{x_1, \ldots, x_m\}$) and the target domain ($D_t = \{x_1, \ldots, x_n\}$), respectively; $c \in \{0, 1\}$ represents the category of solutions, where class 0 and class 1 represent promising solutions and wrong solutions, respectively; $D_s^c$ is the sample set belonging to class $c$ in $D_s$; $D_t^c$ is the sample set belonging to class $c$ in $D_t$; $m_c$ and $n_c$ denote the number of samples in $D_s^c$ and $D_t^c$, respectively; and $\mu \in [0, 1]$ denotes a balance factor that adjusts the importance of the marginal and conditional distributions. In Formula (24), the first and second terms represent the marginal distribution discrepancy and the conditional distribution discrepancy between the domains, respectively.

By further taking advantage of the kernel method and matrix transformation, Formula (24) can be rewritten as:

$$\text{MMD}(D_s, D_t) \;\doteq\; \mathbf{tr}\left( \mathbf{W}^{\mathrm{T}} \mathbf{K} \left( (1 - \mu)\mathbf{M}_0 + \mu \sum_{c=0}^{1} \mathbf{M}_c \right) \mathbf{K}^{\mathrm{T}} \mathbf{W} \right) \tag{25}$$

where $\mathbf{tr}(.)$ refers to the matrix trace; $\mathbf{W}$ denotes the transformation matrix; $\mathbf{K} = \phi(\mathbf{X})^T \phi(\mathbf{X}) \in \mathbb{R}^{(m+n)\times(m+n)}$ represents the kernel matrix; $\mathbf{X}$ denotes the data matrix composed of source and the target domain samples; and $\mathbf{M}_0$ and $\mathbf{M}_c$ are MMD matrices, and their definitions can be found in Wang et al. [13].

Based on Formula (25), finding the optimal feature transformation can be formalised as the following optimisation problem:

$$\begin{aligned} \underset{\mathbf{W}}{\arg\min} \;\; & \text{MMD}(D_s, D_t) + \lambda \cdot \mathbf{tr}(\mathbf{W}^{\mathrm{T}}\mathbf{W}) \\ \text{s.t.} \;\; & \mathbf{W}^{\mathrm{T}}\mathbf{K}\mathbf{H}\mathbf{K}^{\mathrm{T}}\mathbf{W} = \mathbf{I}, \;\; 0 \le \mu \le 1 \end{aligned} \tag{26}$$

In the optimisation problem (26), the second term of the objective function is a regularisation term. The first constraint limits the scatter matrix to maintain the data characteristics of the source and the target domain sample sets. $\mathbf{H}$ is the centring matrix, and $\mathbf{I}$ is the identity matrix.

Problem (26) can be solved by generalised eigenvalue decomposition, and the obtained optimal transformation matrix $\mathbf{W}$ is composed of the first $d$ smallest eigenvectors of the result matrix $\mathbf{R}$. $\mathbf{R}$ is given by Formula (27).

$$\mathbf{R} = \left( \mathbf{K} \left( (1 - \mu)\mathbf{M}_0 + \mu \sum_{c=0}^{1} \mathbf{M}_c \right) \mathbf{K}^{\mathrm{T}} + \lambda \cdot \mathbf{I} \right)^{-1} \mathbf{K}\mathbf{H}\mathbf{K}^{\mathrm{T}} \tag{27}$$

The details of the feature transformation learning are shown in Algorithm 2.

---

**Algorithm 2** K-BDA

---

**Input**: Source and target sample set $D_s$ and $D_t$; kernel function $\kappa$.
**Output**: Feature transformation matrix $\mathbf{W}$.
1: Obtain the class labels of the sampled data in $D_s$ and $D_t$ by using the hierarchical Pareto nondominated sorting method;
2: Construct $\mathbf{K}$, $\mathbf{M}_0$, $\mathbf{M}_c$, $\mathbf{H}$, and $\mathbf{I}$;
3: Build $\mathbf{W}$ by using the $d$ smallest eigenvectors of $\mathbf{R}$;
4: **return W**;

---

Furthermore, we use the knowledge contained in the POF of the source domain to find some solutions expected to perform well in the target domain in the latent space, which is used to construct the initial population of the subsequent evolutionary algorithm. The process of knowledge transfer is as follows: first, the mapping set in the latent space corresponding to the POF of the source domain is calculated. A single-objective evolutionary algorithm is used to find a solution in the target domain for each mapping solution in the mapping set so that it is closest to the mapping solution in the latent space. Next, the initial population is constructed based on the solutions found and the target domain sample set $D_t$. The details of the initial population generation are shown in Algorithm 3.

Algorithm 3 differs from Jiang et al. [46] in two aspects. First, the D-HRDLB problem is a combinatorial optimisation problem with an irregular POF, and the quality of the sample set has a significant impact on subsequent transfer learning. Therefore, different from the latter random sampling, an MOEA-based sampling approach is adopted in our study. Second, the latter does not consider the influence of the conditional distribution of the samples on transfer learning. When the environmental change is small and the source and target domain sample sets are relatively similar, the conditional distribution is more

critical for transfer learning. Therefore, this study considers the marginal and conditional distributions in the sample set simultaneously.

---

**Algorithm 3** B-IPG

---

**Input**: The historical POFs of D-HRDLB problem $POF_0 \ldots, POF_{\ell-1}$; D-HRDLB problem $FC_0 \ldots,$
$FC_\ell$; kernel function $\kappa$.
**Output**: An initial population *init_pop* in the $\ell$th environment ($\ell = 1, 2 \ldots$).
1: Presearch to obtain a sample set $D(\ell)$ of the objective space in the $\ell$th environment;
2: Load the sample sets $D^h = \{D(0) \ldots, D(\ell-1)\}$ of all historical objective spaces;
3: Select a sample set $D(\ell^*) \in D^h$ with the smallest MMD from $D(\ell)$;
4: $D_s \leftarrow D(\ell^*)$; $D_t \leftarrow D(\ell)$; $D^h \leftarrow D^h \cup \{D(\ell)\}$;
5: $\mathbf{W} \leftarrow$ **K-BDA**$(D_s, D_t, \kappa)$;
6: Obtain the set of the mapped samples ($M$) of $POF_{\ell*}$ in the latent space by using $\mathbf{W}$ and $\kappa$;
7: **For** ($m \in M$) **Do**
8:　Compute such a solution $x$ in the objective space in the $\ell$th environment, which is the closest to $m$ in the latent space;
9:　*init_pop* $\leftarrow$ *init_pop* $\cup \{x\}$;
10: **End For**
11: Randomly pick the solutions in $D(\ell)$ to fill *init_pop*;
12: **return** *init_pop*;

---

*5.2. Solution for the D-HRDLB Problem*

5.2.1. Solution Encoding and Decoding

To better adapt to the various existing MOEAs, a real number-based encoding scheme is used to encode solutions to the D-HRDLB problem. The encoding scheme is shown in Figure 5, and the coding sequence in Figure 5 contains five parts. The first part is used to select a feasible disassembly tree for each product at the operation layer of TD-TAOG, with a length of $|\mathbb{B}(\ell)|$. In the first part, $\alpha(B_b)$ refers to the probabilistic priority of each disassembly operation $B_b$ (normal node) to be performed, and its value is set to a real number in the range [0, 1]. The selection process of operations for each product based on the first part (the decoding process) is as follows: for each product, the algorithm traverses all normal nodes and artificial nodes from the artificial root node in the operation layer of its TD-TAOG. For each selected artificial node, the algorithm receives $\alpha(B_b)$ of each normal successor $B_b$ from the first part and selects the normal successor with the largest $\alpha$ and all its artificial successors. This process is iterated towards the leaf nodes to obtain all feasible disassembly trees for all products. Example 1 in Figure 5 shows the feasible disassembly tree for one of the products.

The second part of the coding sequence is used to select a feasible task sequence for each selected disassembly operation $B_b$ (normal node) at the task layer of TD-TAOG. The length of the second part is $|\mathbb{U}(\ell)|$, where $\beta(U_u)$ represents the probabilistic priority of the corresponding task $U_u$ being selected, and its value is set to an actual number in the range [0, 1]. The task selection process based on the second part is as follows: for each selected disassembly operation, the algorithm traverses all feasible unit nodes in the current environment from the initial node in the corresponding task graph. For each feasible unit node selected, the algorithm obtains $\beta(U_u)$ of each succeeding feasible unit node $U_u$ from the second part and selects the feasible unit node with the largest $\beta$. This process is iterated towards the terminal nodes to obtain feasible task sequences for all selected disassembly operations. Example 2 in Figure 5 shows the feasible task sequence $[U_2, U_5, U_7]$ required to complete the selected disassembly operation $B_9$.

The third part of the coding sequence determines the correspondence between workstations and operators, i.e., if and only if $\lfloor \gamma(O_o) \cdot |\mathbb{W}| \rfloor = O$, where $\gamma(O_o)$ refers to a real number in the range [0, 1]. The length of the third part is $|\mathbb{R}| + |\mathbb{H}|$. For each idle operator not assigned to the workstation, its corresponding $\gamma$ in the third part is set to 1. In example 3 in Figure 5, the third part shows that $O_1$, $O_4$, $O_7$, and $O_9$ should be assigned to workstation $W_1$.

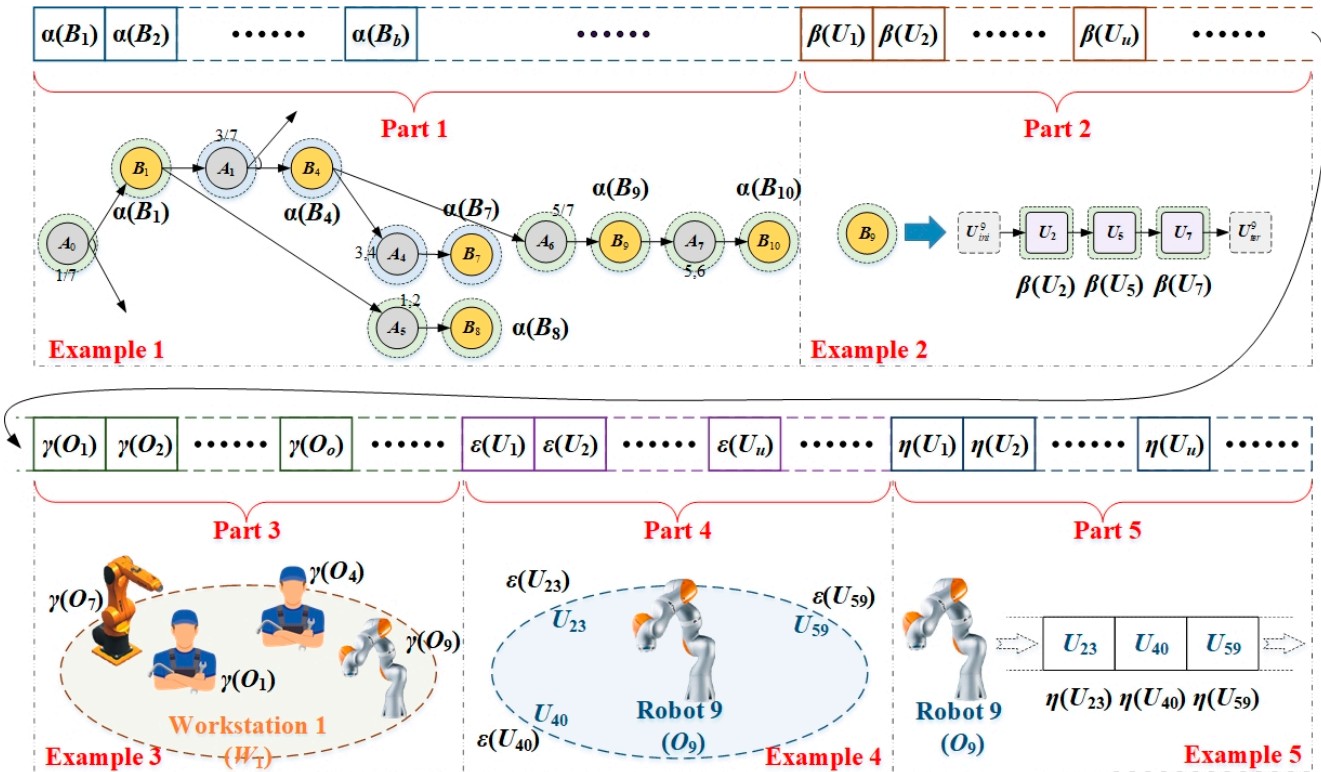

**Figure 5.** The encoding scheme for solving the D-HRDLB problem.

The fourth part of the coding sequence determines the correspondence between the selected tasks and operators. Its length is $|\mathbb{U}(\ell)|$, and $\varepsilon(U_u)$ is a real number in the range between [0, 1). If task $U_u$ is selected, it will be assigned to the operator with index $\lfloor \varepsilon(U_u) \cdot (|\mathbb{R}| + |\mathbb{H}|) \rfloor$ to perform, and the operator must be assigned to a workstation. In example 4 in Figure 5, the fourth part shows that $U_{23}$, $U_{40}$, and $U_{59}$ should be assigned to the ninth operator (robot 9) to perform.

The last part of the coding sequence determines the probabilistic priority of tasks that are performed by the same operator and have no precedence constraint between them. The length of the last part is $|\mathbb{U}(\ell)|$. $\eta(U_u)$ represents the probabilistic priority of the corresponding task $U_u$, and its value is set to a real number in the range [0, 1]. In example 5 in Figure 5, $U_{23}$, $U_{40}$, and $U_{59}$ are assigned to robot 9 to perform, and there is no precedence constraint between them. Since $\eta(U_{23}) > \eta(U_{40}) > \eta(U_{59})$, robot 9 will perform $U_{23}$ first, then $U_{40}$, and finally $U_{59}$.

### 5.2.2. Solution Initialisation and Variation

Due to the real number-based encoding scheme, the solution initialisation can start from the random sampling of each element of the coding sequence in its corresponding range. The crossover and mutation operators also use the classic simulated binary crossover (SBX) and polynomial mutation (PM) [48], respectively. However, the obtained solutions are likely to be infeasible due to the close connection between the various parts of the coding sequence and multiple constraints, such as precedence constraints, in the D-HRDLB problem. Therefore, solution initialisation and variation need to follow some rules and perform some repair operations to ensure the feasibility of solutions.

(1)  The values $\alpha$ of all succeeding normal nodes of each artificial node in a TD-TAOG are not equal.
(2)  The values $\beta$ of all succeeding unit nodes in the same state of each initial node or unit node are not equal.

(3) For any disassembly operation, at least one feasible task sequence can complete it in any state.

(4) The range between the number of operators assigned to any workstation is [1, *TO*].

(5) Each selected task can only be assigned to one operator in its feasible operation.

(6) Operators needed to perform tasks must be assigned to workstations.

(7) All selected tasks belonging to the same disassembly operation should be performed in the same workstation, and the operators performing these tasks should be assigned to the same workstation.

(8) For any two selected tasks $U_u$ and $U_{u'}$, if the disassembly operation to which $U_u$ belongs is the predecessor of the disassembly operation to which $U_{u'}$ belongs, the operator performing $U_u$ is either at the same workstation as the operator performing $U_{u'}$ or at the previous workstation of the workstation where the operator who performs $U_{u'}$ is located.

(9) The values $\eta$ of the tasks performed by the same operator without precedence constraint are not equal.

## 6. Computational Experiments

### 6.1. Experimental Settings

The TD-TAOG and the D-HRDLB problem are studied for the first time in this paper, so it is necessary to generate new problem instances for algorithm comparison. By adding a task graph for each normal node in the TAOGs of six realistic EOL products from the literature, six new TD-TAOGs are generated. They are named TD-TAOG 1 [7], TD-TAOG 2 [49], TD-TAOG 3 [45], TD-TAOG 4 [50], TD-TAOG 5 [51], and TD-TAOG 6 [51]. These six products are disassembled on a human-robot collaborative disassembly line. The number of workstations is set to three, and the number of optional robots and humans is set to 24 and 10, respectively. The range of the number of robots and humans that can be allocated in each workstation is [1, 4] and [1, 3], respectively. The processing times of tasks by robots and humans are randomly generated in [0.7·$t_{ub}$, 1.3·$t_{ub}$] and [0.4·$t_{ub}$, 0.6·$t_{ub}$], respectively, where $t_{ub}$ is the referenced task processing time.

In this section, we build dynamic disassembly environments based on the first uncertainty, which is the uncertain product quality. Specifically, uncertain product quality may change the states of some disassembly operations, which may cause the corresponding task sequences to be infeasible. Therefore, it is necessary to reselect feasible task sequences for these operations. Table 4 describes this dynamic factor. In Table 4, the disassembly operations that may be in a damaged or missing state in each TD-TAOG are given. For example, in the first environment ($\ell = 0$), disassembly operation nine in TD-TAOG 1 is damaged, while operations 20 and 21 are missing, and the remaining operations are in a normal state. In the second environment ($\ell = 1$), operation nine in TD-TAOG 1 is missing, while operations 21 and 23 are damaged, and the rest of the operations are normal. We set up eight different environments ($\ell = 0, \ldots, 7$) for each D-HRDLB problem instance. The disassembly operations in a damaged or missing state in each environment are randomly determined based on Table 4. A set of D-HRDLB problem instances, including small-scale and large-scale instances, are generated based on the above settings. In the set, 15 small-scale instances are composed of any two of the six TD-TAOGs, and 24 large-scale instances are composed of any three to four TD-TAOGs. The details of the 39 instances are shown in Table 5.

The proposed B-DMOEA is an evolutionary dynamic optimisation algorithm that is assisted by feature-based transfer learning. In fact, any population-based metaheuristic, including evolutionary approaches, such as genetic algorithms and swarm intelligence-based methods (including particle swarm optimisation algorithms), can be embedded in the B-DMOEA. However, this study mainly focuses on the design of the evolutionary dynamic optimisation method. Therefore, three representative MOEAs, including NSGA-II [48], RVEA-iGNG [52], and IBEA [53], are selected, leading the evolution via the Pareto criterion, decomposition-based criterion, and indicator-based criterion, respectively. These

three MOEAs are embedded in the Tr-DMOEA proposed by Jiang et al. [46] and our B-DMOEA to analyse the performance difference between B-DMOEA and Tr-DMOEA in solving the D-HRDLB problem. The three algorithms embedded in Tr-DMOEA are called Tr-NSGA-II, Tr-REVA-i, and Tr-IBEA, respectively, and the three algorithms embedded in B-DMOEA are called B-NSGA-II, B-REVA-i, and B-IBEA, respectively. It should be noted that few evolutionary dynamic optimisation algorithms are assisted by feature-based transfer learning, such as Tr-DMOEA and B-DMOEA. Therefore, Tr-DMOEA is chosen as the competitor of B-DMOEA in this study.

**Table 4.** Disassembly operations that may be in the damaged or missing state.

| Models | Uncertain Disassembly Operations |
|---|---|
| TD-TAOG 1 | 9 |
| TD-TAOG 2 | 20, 21, 22, 23 |
| TD-TAOG 3 | 6, 11 |
| TD-TAOG 4 | 7, 11 |
| TD-TAOG 5 | 9, 10, 11 |
| TD-TAOG 6 | 6, 7, 17, 18, 19, 30 |

**Table 5.** Details of the 39 instances.

| Problem | Scale | Products | Tasks | Problem | Scale | Products | Tasks |
|---|---|---|---|---|---|---|---|
| S1 | Small | 1, 2 | 241 | L6 | Large | 1, 3, 5 | 375 |
| S2 | Small | 1, 3 | 217 | L7 | Large | 1, 3, 6 | 435 |
| S3 | Small | 1, 4 | 164 | L8 | Large | 1, 4, 5 | 322 |
| S4 | Small | 1, 5 | 232 | L9 | Large | 1, 4, 6 | 382 |
| S5 | Small | 1, 6 | 292 | L10 | Large | 1, 5, 6 | 450 |
| S6 | Small | 2, 3 | 310 | L11 | Large | 2, 3, 4 | 400 |
| S7 | Small | 2, 4 | 257 | L12 | Large | 2, 3, 5 | 468 |
| S8 | Small | 2, 5 | 325 | L13 | Large | 2, 3, 6 | 528 |
| S9 | Small | 2, 6 | 385 | L14 | Large | 2, 4, 5 | 415 |
| S10 | Small | 3, 4 | 233 | L15 | Large | 2, 4, 6 | 475 |
| S11 | Small | 3, 5 | 301 | L16 | Large | 2, 5, 6 | 543 |
| S12 | Small | 3, 6 | 361 | L17 | Large | 3, 4, 5 | 391 |
| S13 | Small | 4, 5 | 248 | L18 | Large | 3, 4, 6 | 451 |
| S14 | Small | 4, 6 | 308 | L19 | Large | 3, 5, 6 | 519 |
| S15 | Small | 5, 6 | 376 | L20 | Large | 4, 5, 6 | 466 |
| L1 | Large | 1, 2, 3 | 384 | L21 | Large | 1, 2, 3, 4 | 474 |
| L2 | Large | 1, 2, 4 | 331 | L22 | Large | 1, 2, 3, 5 | 542 |
| L3 | Large | 1, 2, 5 | 399 | L23 | Large | 1, 2, 4, 5 | 489 |
| L4 | Large | 1, 2, 6 | 459 | L24 | Large | 1, 2, 4, 6 | 549 |
| L5 | Large | 1, 3, 4 | 307 | | | | |

Our previous work shows that the POFs of combinatorial optimisation problems, such as the D-HRDLB problem, are generally irregular [5]. Traditional decomposition-based evolutionary algorithms, such as MOEA/D [54], assume that the distribution of reference vectors can be predefined. However, the distribution of reference vectors may not match the irregular POFs, resulting in wasted vectors that have no reasonable solutions nearby and bad vectors that may bias the search direction. Therefore, we choose the adaptive reference vector-guided RVEA-iGNG instead of MOEA/D as a representative algorithm for decomposition-based MOEAs. In addition, the fine performance of NSGA-II and IBEA in solving DLBPs has been proven by Fang et al. [5]. Consequently, choosing these three MOEAs can strengthen the confidence level of the experimental results.

In this paper, MIGD and MHV are used to quantify the performance of B-DMOEA. They are variants of IGD [55] and HV [53] for dynamic multiobjective optimisation algorithms. MIGD is defined as the average value of the IGD values in all environments [46]. MHV is defined as the average value of the HV values in all environments. Note that the exact Pareto font of the D-HRDLB problem is unknown. Therefore, we use a collection of all the solutions, which we obtained by executing all the runs for all the algorithms and picking the nondominated solutions from the collection as the ideal solution set to obtain the approximate Pareto front. We normalise the solution set obtained by each algorithm

with the approximate Pareto front and calculate the corresponding MIGD and MHV based on these normalised values. The reference point is set to 1.1 times the nadir point of the approximate Pareto front.

For each problem instance, each algorithm is executed with ten independent runs to obtain the mean and standard deviation values of each metric that evaluates the algorithm. The size of the population in all the algorithms is set to 150. In each environment, the presearch, single-objective search, and multiobjective search are repeated for 3, 3, and 30 generations, respectively. The kernel function in Tr-DMOEA and B-DMOEA selects the linear kernel function, while the dimensionality is set to 100, and the weight λ of the regularisation term is set to 0.1. Other parameters are set according to the suggestions given by the developers.

### 6.2. Experimental Results and Discussion

#### 6.2.1. Parameter Sensitivity

The value of μ in the B-DMOEA determines the importance of the marginal and conditional distributions in the knowledge transfer. In this section, we analyse the relationship between μ and problem instances to guide the setting of the optimal value of μ. We select three problem instances of different scales and analyse the MIGD and MHV of B-NSGA-II, B-REVA-i and the B-IBEA on these three problem instances when μ takes different values (μ ∈ {0, 0.1, . . . , 1.0}). The experimental results are shown in Figure 6.

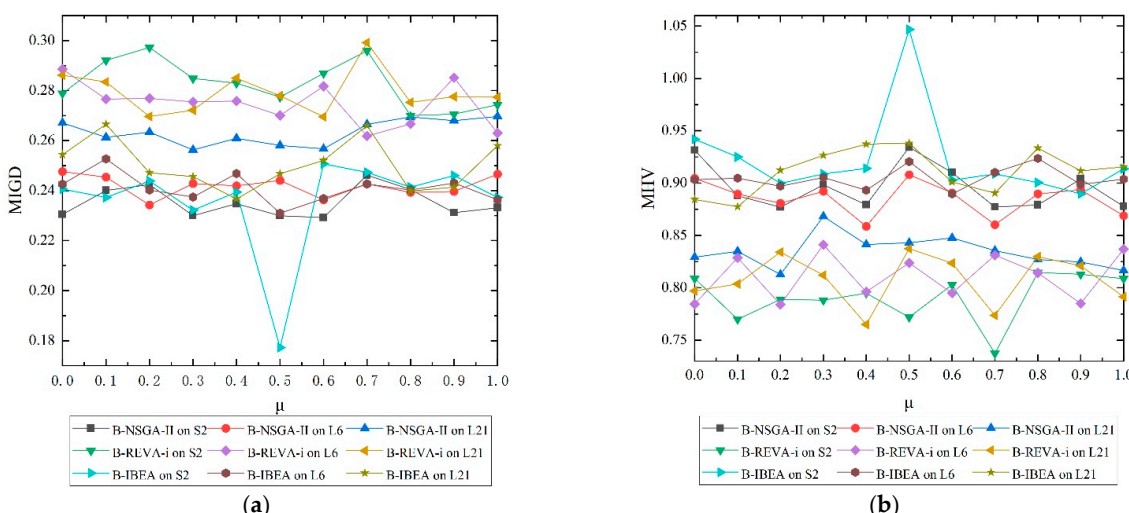

**Figure 6.** MIGD and MHV in relation to μ on different problem instances. (**a**) MIGD obtained by all algorithms; (**b**) MHV obtained by all algorithms.

Figure 6 shows that when μ = 0, that is, when only the marginal distributions are considered, the three algorithms cannot obtain the optimal MIGD and MHV values on the three problem instances, indicating the importance of considering both marginal and conditional distributions in our B-DMOEA. Moreover, Figure 6 shows that the optimal μ varies on different problem instances. For example, in Figure 6b, when μ = 0.5, B-NSGA-II can obtain the optimal MHV on both problem instances S2 and L6, which shows that marginal and conditional distributions contribute equally to the discrepancy. In contrast, when μ = 0.3, B-NSGA-II can obtain the optimal MHV on the problem instance L21, which means the marginal distributions contribute more to the discrepancy than the conditional distributions, and the performance of transfer learning depends more on the marginal distributions. The observations indicate the importance of balancing the marginal and conditional distributions based on the characteristics of the problem instances. In the other subfigures in Figure 6, the observations are similar. Therefore, B-DMOEA is more likely to achieve a better performance than Tr-DMOEA. Note that Figure 6 indicates that the values of μ need to be adjusted for problem instances. However, the experiments in this

paper mainly aim to prove the importance of both marginal and conditional distributions between domains in feature-based transfer learning approaches. Moreover, it can also be seen from Figure 6 that when μ = 0.5, B-DMOEA can perform better in most cases. Therefore, in subsequent experiments, μ is uniformly set to 0.5 without adjusting to different problem instances.

In addition, Figure 6 shows that decomposition-based algorithms, such as B-REVA-I, are more unstable than B-NSGA-II and B-IBEA, proving that decomposition-based algorithms have difficulty solving real-world problems (e.g., D-HRDLB) in which the Pareto front geometry is often irregular. This finding is consistent with the results of our previous work [5].

### 6.2.2. Performance Evaluation of B-DMOEA

This section analyses the performance differences between the B-DMOEA and each competitor in solving the D-HRDLB problem. Tables 6 and 7 provide the MIGD results (mean and standard deviation) for the three groups of paired algorithms, including Tr-NSGA-II versus B-NSGA-II, Tr-RVEA-i versus B-RVEA-i, and Tr-IBEA versus B-IBEA, on small-scale and large-scale problem instances. The better mean for each instance is highlighted in boldface.

**Table 6.** Mean and standard deviation values of the MIGD metric obtained by comparing algorithms for small-scale instances.

| Problems | Tr-NSGA-II | B-NSGA-II | Tr-RVEA-i | B-RVEA-i | Tr-IBEA | B-IBEA |
|---|---|---|---|---|---|---|
| S1 | $2.4452 \times 10^{-1}$ [$1.4 \times 10^{-2}$] | **$1.8438 \times 10^{-1}$ [$1.4 \times 10^{-2}$]** | $2.9134 \times 10^{-1}$ [$3.0 \times 10^{-2}$] | **$2.8346 \times 10^{-1}$ [$3.5 \times 10^{-2}$]** | **$2.4248 \times 10^{-1}$ [$2.7 \times 10^{-2}$]** | $2.6209 \times 10^{-1}$ [$1.9 \times 10^{-2}$] |
| S2 | $2.3724 \times 10^{-1}$ [$1.0 \times 10^{-2}$] | **$2.2995 \times 10^{-1}$ [$1.9 \times 10^{-2}$]** | $3.0424 \times 10^{-1}$ [$3.2 \times 10^{-2}$] | **$2.7740 \times 10^{-1}$ [$2.4 \times 10^{-2}$]** | $2.4383 \times 10^{-1}$ [$2.1 \times 10^{-2}$] | **$1.7722 \times 10^{-1}$ [$1.1 \times 10^{-2}$]** |
| S3 | $2.2963 \times 10^{-1}$ [$2.4 \times 10^{-2}$] | **$2.1814 \times 10^{-1}$ [$1.1 \times 10^{-2}$]** | $3.0019 \times 10^{-1}$ [$2.7 \times 10^{-2}$] | $3.0060 \times 10^{-1}$ [$3.3 \times 10^{-2}$] | $2.3418 \times 10^{-1}$ [$3.8 \times 10^{-2}$] | **$2.2536 \times 10^{-1}$ [$2.1 \times 10^{-2}$]** |
| S4 | **$2.1752 \times 10^{-1}$ [$1.1 \times 10^{-2}$]** | $2.1784 \times 10^{-1}$ [$1.3 \times 10^{-2}$] | **$2.7443 \times 10^{-1}$ [$2.8 \times 10^{-2}$]** | $2.8932 \times 10^{-1}$ [$3.5 \times 10^{-2}$] | $2.3339 \times 10^{-1}$ [$1.7 \times 10^{-2}$] | **$2.3077 \times 10^{-1}$ [$1.8 \times 10^{-2}$]** |
| S5 | $2.3844 \times 10^{-1}$ [$1.1 \times 10^{-2}$] | **$2.3323 \times 10^{-1}$ [$1.7 \times 10^{-2}$]** | $2.9138 \times 10^{-1}$ [$2.3 \times 10^{-2}$] | **$2.8994 \times 10^{-1}$ [$1.4 \times 10^{-2}$]** | $2.6316 \times 10^{-1}$ [$3.1 \times 10^{-2}$] | **$2.5113 \times 10^{-1}$ [$4.9 \times 10^{-2}$]** |
| S6 | $2.3010 \times 10^{-1}$ [$1.9 \times 10^{-2}$] | **$2.1976 \times 10^{-1}$ [$1.4 \times 10^{-2}$]** | $2.8260 \times 10^{-1}$ [$3.2 \times 10^{-2}$] | **$2.5600 \times 10^{-1}$ [$2.2 \times 10^{-2}$]** | $2.3648 \times 10^{-1}$ [$1.7 \times 10^{-2}$] | **$1.8782 \times 10^{-1}$ [$9.1 \times 10^{-3}$]** |
| S7 | $2.2801 \times 10^{-1}$ [$2.3 \times 10^{-2}$] | **$2.1860 \times 10^{-1}$ [$1.3 \times 10^{-2}$]** | $2.7257 \times 10^{-1}$ [$2.40 \times 10^{-2}$] | **$2.5558 \times 10^{-1}$ [$2.3 \times 10^{-2}$]** | $2.3089 \times 10^{-1}$ [$1.4 \times 10^{-2}$] | **$2.1843 \times 10^{-1}$ [$1.6 \times 10^{-2}$]** |
| S8 | $1.9563 \times 10^{-1}$ [$5.8 \times 10^{-3}$] | **$1.8677 \times 10^{-1}$ [$1.8 \times 10^{-2}$]** | $2.5834 \times 10^{-1}$ [$2.2 \times 10^{-2}$] | **$2.5619 \times 10^{-1}$ [$2.2 \times 10^{-2}$]** | $1.9090 \times 10^{-1}$ [$1.1 \times 10^{-2}$] | **$1.8166 \times 10^{-1}$ [$7.5 \times 10^{-3}$]** |
| S9 | $2.3783 \times 10^{-1}$ [$2.2 \times 10^{-2}$] | **$2.2976 \times 10^{-1}$ [$1.3 \times 10^{-2}$]** | **$2.8184 \times 10^{-1}$ [$2.1 \times 10^{-2}$]** | $2.8756 \times 10^{-1}$ [$1.7 \times 10^{-2}$] | $2.4074 \times 10^{-1}$ [$1.4 \times 10^{-2}$] | **$2.2255 \times 10^{-1}$ [$1.7 \times 10^{-2}$]** |
| S10 | $2.4093 \times 10^{-1}$ [$1.4 \times 10^{-2}$] | **$2.2855 \times 10^{-1}$ [$1.4 \times 10^{-2}$]** | **$2.8324 \times 10^{-1}$ [$1.8 \times 10^{-2}$]** | $2.9758 \times 10^{-1}$ [$2.2 \times 10^{-2}$] | $2.4034 \times 10^{-1}$ [$1.8 \times 10^{-2}$] | **$2.3150 \times 10^{-1}$ [$1.1 \times 10^{-2}$]** |
| S11 | **$2.1666 \times 10^{-1}$ [$1.7 \times 10^{-2}$]** | $2.2573 \times 10^{-1}$ [$1.3 \times 10^{-2}$] | $2.7445 \times 10^{-1}$ [$1.6 \times 10^{-2}$] | **$2.6756 \times 10^{-1}$ [$1.6 \times 10^{-2}$]** | $2.2533 \times 10^{-1}$ [$2.1 \times 10^{-2}$] | **$2.2233 \times 10^{-1}$ [$2.1 \times 10^{-2}$]** |
| S12 | $2.5712 \times 10^{-1}$ [$1.8 \times 10^{-2}$] | **$2.3898 \times 10^{-1}$ [$1.5 \times 10^{-2}$]** | **$2.8810 \times 10^{-1}$ [$3.20 \times 10^{-2}$]** | $2.9191 \times 10^{-1}$ [$2.6 \times 10^{-2}$] | $2.4306 \times 10^{-1}$ [$2.6 \times 10^{-2}$] | **$2.3886 \times 10^{-1}$ [$1.5 \times 10^{-2}$]** |
| S13 | $2.2702 \times 10^{-1}$ [$1.8 \times 10^{-2}$] | **$2.1672 \times 10^{-1}$ [$1.1 \times 10^{-2}$]** | $2.7059 \times 10^{-1}$ [$1.9 \times 10^{-2}$] | **$2.5719 \times 10^{-1}$ [$1.8 \times 10^{-2}$]** | $2.3363 \times 10^{-1}$ [$1.7 \times 10^{-2}$] | **$2.2914 \times 10^{-1}$ [$2.1 \times 10^{-2}$]** |
| S14 | $2.4190 \times 10^{-1}$ [$1.6 \times 10^{-2}$] | **$2.1207 \times 10^{-1}$ [$1.2 \times 10^{-2}$]** | **$2.9200 \times 10^{-1}$ [$2.4 \times 10^{-2}$]** | $2.9462 \times 10^{-1}$ [$1.8 \times 10^{-2}$] | $2.4505 \times 10^{-1}$ [$2.8 \times 10^{-2}$] | **$2.1807 \times 10^{-1}$ [$2.4 \times 10^{-2}$]** |
| S15 | $2.5624 \times 10^{-1}$ [$2.0 \times 10^{-2}$] | **$2.4114 \times 10^{-1}$ [$1.5 \times 10^{-2}$]** | $3.1559 \times 10^{-1}$ [$2.4 \times 10^{-2}$] | **$2.9914 \times 10^{-1}$ [$2.7 \times 10^{-2}$]** | $2.3386 \times 10^{-1}$ [$1.9 \times 10^{-2}$] | **$2.3033 \times 10^{-1}$ [$2.7 \times 10^{-2}$]** |

The better mean for each instance is highlighted in boldface.

**Table 7.** Mean and standard deviation values of the MIGD metric obtained by comparing algorithms for large-scale instances.

| Problems | Tr-NSGA-II | B-NSGA-II | Tr-RVEA-i | B-RVEA-i | Tr-IBEA | B-IBEA |
|---|---|---|---|---|---|---|
| L1 | $2.3935 \times 10^{-1}$ [$1.2 \times 10^{-2}$] | **$2.3063 \times 10^{-1}$ [$6.2 \times 10^{-3}$]** | $2.9619 \times 10^{-1}$ [$3.0 \times 10^{-2}$] | **$2.7113 \times 10^{-1}$ [$3.4 \times 10^{-2}$]** | $2.5128 \times 10^{-1}$ [$2.6 \times 10^{-2}$] | **$2.4689 \times 10^{-1}$ [$1.9 \times 10^{-2}$]** |
| L2 | **$2.3537 \times 10^{-1}$ [$1.8 \times 10^{-2}$]** | $2.3924 \times 10^{-1}$ [$2.4 \times 10^{-2}$] | $2.8540 \times 10^{-1}$ [$3.5 \times 10^{-2}$] | **$2.7994 \times 10^{-1}$ [$2.5 \times 10^{-2}$]** | $2.5117 \times 10^{-1}$ [$3.8 \times 10^{-2}$] | **$2.3892 \times 10^{-1}$ [$2.0 \times 10^{-2}$]** |
| L3 | **$2.3213 \times 10^{-1}$ [$1.8 \times 10^{-2}$]** | $2.4014 \times 10^{-1}$ [$2.4 \times 10^{-2}$] | $2.8749 \times 10^{-1}$ [$3.5 \times 10^{-2}$] | **$2.7015 \times 10^{-1}$ [$2.5 \times 10^{-2}$]** | **$2.4205 \times 10^{-1}$ [$3.8 \times 10^{-2}$]** | $2.6396 \times 10^{-1}$ [$2.0 \times 10^{-2}$] |
| L4 | $2.4273 \times 10^{-1}$ [$1.3 \times 10^{-2}$] | **$2.3039 \times 10^{-1}$ [$2.0 \times 10^{-2}$]** | **$2.8185 \times 10^{-1}$ [$2.2 \times 10^{-2}$]** | $2.9387 \times 10^{-1}$ [$4.2 \times 10^{-2}$] | $2.6632 \times 10^{-1}$ [$2.8 \times 10^{-2}$] | **$2.4615 \times 10^{-1}$ [$3.2 \times 10^{-2}$]** |
| L5 | $2.4366 \times 10^{-1}$ [$1.9 \times 10^{-2}$] | **$2.3203 \times 10^{-1}$ [$1.9 \times 10^{-2}$]** | $2.7817 \times 10^{-1}$ [$2.0 \times 10^{-2}$] | **$2.7550 \times 10^{-1}$ [$1.8 \times 10^{-2}$]** | **$2.3752 \times 10^{-1}$ [$2.8 \times 10^{-2}$]** | $2.4015 \times 10^{-1}$ [$1.5 \times 10^{-2}$] |
| L6 | $2.4881 \times 10^{-1}$ [$2.5 \times 10^{-2}$] | **$2.4405 \times 10^{-1}$ [$1.3 \times 10^{-2}$]** | $2.7308 \times 10^{-1}$ [$2.8 \times 10^{-2}$] | **$2.7005 \times 10^{-1}$ [$1.3 \times 10^{-2}$]** | $2.4742 \times 10^{-1}$ [$2.0 \times 10^{-2}$] | **$2.3097 \times 10^{-1}$ [$1.8 \times 10^{-2}$]** |
| L7 | $2.6642 \times 10^{-1}$ [$2.8 \times 10^{-2}$] | **$2.5105 \times 10^{-1}$ [$1.6 \times 10^{-2}$]** | $2.8990 \times 10^{-1}$ [$3.2 \times 10^{-2}$] | **$2.8106 \times 10^{-1}$ [$2.5 \times 10^{-2}$]** | $2.5246 \times 10^{-1}$ [$1.5 \times 10^{-2}$] | **$2.4094 \times 10^{-1}$ [$1.8 \times 10^{-2}$]** |
| L8 | $2.3601 \times 10^{-1}$ [$1.9 \times 10^{-2}$] | **$2.2650 \times 10^{-1}$ [$1.4 \times 10^{-2}$]** | **$2.5279 \times 10^{-1}$ [$2.1 \times 10^{-2}$]** | $2.6524 \times 10^{-1}$ [$3.2 \times 10^{-2}$] | $2.4003 \times 10^{-1}$ [$2.9 \times 10^{-2}$] | **$2.3166 \times 10^{-1}$ [$2.4 \times 10^{-2}$]** |
| L9 | $2.4765 \times 10^{-1}$ [$2.8 \times 10^{-2}$] | **$2.4503 \times 10^{-1}$ [$3.2 \times 10^{-2}$]** | $2.8767 \times 10^{-1}$ [$3.1 \times 10^{-2}$] | **$2.7025 \times 10^{-1}$ [$2.2 \times 10^{-2}$]** | **$2.6361 \times 10^{-1}$ [$3.2 \times 10^{-2}$]** | $2.6835 \times 10^{-1}$ [$4.9 \times 10^{-2}$] |
| L10 | $2.7371 \times 10^{-1}$ [$2.8 \times 10^{-2}$] | **$2.5823 \times 10^{-1}$ [$2.2 \times 10^{-2}$]** | $2.7792 \times 10^{-1}$ [$2.2 \times 10^{-2}$] | **$2.7048 \times 10^{-1}$ [$2.5 \times 10^{-2}$]** | $2.6016 \times 10^{-1}$ [$2.2 \times 10^{-2}$] | **$2.5412 \times 10^{-1}$ [$2.0 \times 10^{-2}$]** |
| L11 | **$2.3321 \times 10^{-1}$ [$1.9 \times 10^{-2}$]** | $2.4112 \times 10^{-1}$ [$3.2 \times 10^{-2}$] | **$2.6612 \times 10^{-1}$ [$1.2 \times 10^{-2}$]** | $2.7659 \times 10^{-1}$ [$1.6 \times 10^{-2}$] | **$2.3368 \times 10^{-1}$ [$3.0 \times 10^{-2}$]** | $2.3919 \times 10^{-1}$ [$3.0 \times 10^{-2}$] |
| L12 | $2.4280 \times 10^{-1}$ [$1.7 \times 10^{-2}$] | **$2.3355 \times 10^{-1}$ [$1.6 \times 10^{-2}$]** | $2.8230 \times 10^{-1}$ [$3.1 \times 10^{-2}$] | **$2.7953 \times 10^{-1}$ [$1.2 \times 10^{-2}$]** | $2.5248 \times 10^{-1}$ [$1.3 \times 10^{-2}$] | **$2.4538 \times 10^{-1}$ [$3.1 \times 10^{-2}$]** |
| L13 | **$2.2932 \times 10^{-1}$ [$2.6 \times 10^{-2}$]** | $2.4035 \times 10^{-1}$ [$1.5 \times 10^{-2}$] | $2.7526 \times 10^{-1}$ [$3.1 \times 10^{-2}$] | **$2.7454 \times 10^{-1}$ [$2.4 \times 10^{-2}$]** | **$2.2399 \times 10^{-1}$ [$1.9 \times 10^{-2}$]** | $2.2590 \times 10^{-1}$ [$1.4 \times 10^{-2}$] |
| L14 | $2.3022 \times 10^{-1}$ [$2.7 \times 10^{-2}$] | **$2.2218 \times 10^{-1}$ [$1.3 \times 10^{-2}$]** | $2.7224 \times 10^{-1}$ [$2.2 \times 10^{-2}$] | **$2.6957 \times 10^{-1}$ [$2.2 \times 10^{-2}$]** | $2.4100 \times 10^{-1}$ [$2.9 \times 10^{-2}$] | **$2.2163 \times 10^{-1}$ [$2.0 \times 10^{-2}$]** |
| L15 | **$2.3626 \times 10^{-1}$ [$2.0 \times 10^{-2}$]** | $2.4052 \times 10^{-1}$ [$2.9 \times 10^{-2}$] | $2.9041 \times 10^{-1}$ [$2.4 \times 10^{-2}$] | **$2.8585 \times 10^{-1}$ [$1.9 \times 10^{-2}$]** | **$2.3801 \times 10^{-1}$ [$1.3 \times 10^{-2}$]** | $2.4372 \times 10^{-1}$ [$2.4 \times 10^{-2}$] |
| L16 | $2.4780 \times 10^{-1}$ [$2.7 \times 10^{-2}$] | **$2.3091 \times 10^{-1}$ [$1.6 \times 10^{-2}$]** | $2.8216 \times 10^{-1}$ [$2.7 \times 10^{-2}$] | **$2.7861 \times 10^{-1}$ [$1.8 \times 10^{-2}$]** | **$2.4878 \times 10^{-1}$ [$1.5 \times 10^{-2}$]** | $2.5932 \times 10^{-1}$ [$4.0 \times 10^{-2}$] |
| L17 | $2.2096 \times 10^{-1}$ [$2.4 \times 10^{-2}$] | **$2.2051 \times 10^{-1}$ [$1.7 \times 10^{-2}$]** | **$2.5461 \times 10^{-1}$ [$2.5 \times 10^{-1}$]** | $2.6527 \times 10^{-1}$ [$1.3 \times 10^{-2}$] | **$2.2753 \times 10^{-1}$ [$2.3 \times 10^{-1}$]** | $2.3158 \times 10^{-1}$ [$2.3 \times 10^{-1}$] |
| L18 | $2.3247 \times 10^{-1}$ [$2.7 \times 10^{-2}$] | **$2.3100 \times 10^{-1}$ [$1.2 \times 10^{-2}$]** | $2.7617 \times 10^{-1}$ [$1.9 \times 10^{-2}$] | **$2.7364 \times 10^{-1}$ [$2.3 \times 10^{-2}$]** | $2.4562 \times 10^{-1}$ [$2.2 \times 10^{-2}$] | **$2.2186 \times 10^{-1}$ [$1.6 \times 10^{-2}$]** |
| L19 | **$2.3837 \times 10^{-1}$ [$2.4 \times 10^{-2}$]** | $2.4017 \times 10^{-1}$ [$2.1 \times 10^{-2}$] | **$2.8926 \times 10^{-1}$ [$2.6 \times 10^{-2}$]** | $2.8989 \times 10^{-1}$ [$3.2 \times 10^{-2}$] | **$2.4037 \times 10^{-1}$ [$2.3 \times 10^{-2}$]** | $2.5577 \times 10^{-1}$ [$3.8 \times 10^{-2}$] |
| L20 | $2.5089 \times 10^{-1}$ [$2.0 \times 10^{-2}$] | **$2.4525 \times 10^{-1}$ [$1.1 \times 10^{-2}$]** | **$2.8957 \times 10^{-1}$ [$2.5 \times 10^{-2}$]** | $2.9935 \times 10^{-1}$ [$2.8 \times 10^{-2}$] | $2.6209 \times 10^{-1}$ [$4.3 \times 10^{-2}$] | **$2.5611 \times 10^{-1}$ [$3.5 \times 10^{-2}$]** |
| L21 | $2.4511 \times 10^{-1}$ [$2.3 \times 10^{-2}$] | **$2.4499 \times 10^{-1}$ [$2.1 \times 10^{-2}$]** | $2.6704 \times 10^{-1}$ [$1.8 \times 10^{-2}$] | **$2.5905 \times 10^{-1}$ [$2.1 \times 10^{-2}$]** | $2.4263 \times 10^{-1}$ [$1.6 \times 10^{-2}$] | **$2.3891 \times 10^{-1}$ [$2.0 \times 10^{-2}$]** |
| L22 | $2.7039 \times 10^{-1}$ [$2.8 \times 10^{-2}$] | **$2.5806 \times 10^{-1}$ [$1.4 \times 10^{-2}$]** | **$2.7736 \times 10^{-1}$ [$2.3 \times 10^{-2}$]** | $2.7794 \times 10^{-1}$ [$2.0 \times 10^{-2}$] | $2.5850 \times 10^{-1}$ [$2.0 \times 10^{-2}$] | **$2.4672 \times 10^{-1}$ [$2.8 \times 10^{-2}$]** |
| L23 | $2.5855 \times 10^{-1}$ [$1.3 \times 10^{-2}$] | **$2.4295 \times 10^{-1}$ [$2.0 \times 10^{-2}$]** | $2.7430 \times 10^{-1}$ [$1.9 \times 10^{-2}$] | **$2.6308 \times 10^{-1}$ [$2.0 \times 10^{-2}$]** | $2.4715 \times 10^{-1}$ [$1.2 \times 10^{-2}$] | **$2.3413 \times 10^{-1}$ [$1.4 \times 10^{-2}$]** |
| L24 | $2.4900 \times 10^{-1}$ [$1.5 \times 10^{-2}$] | **$2.4247 \times 10^{-1}$ [$2.4 \times 10^{-2}$]** | **$2.7597 \times 10^{-1}$ [$1.8 \times 10^{-2}$]** | $2.9069 \times 10^{-1}$ [$1.9 \times 10^{-2}$] | **$2.4801 \times 10^{-1}$ [$2.7 \times 10^{-2}$]** | $2.4963 \times 10^{-1}$ [$2.8 \times 10^{-2}$] |

The better mean for each instance is highlighted in boldface.

As shown in Tables 6 and 7, the three B-DMOEA algorithms generally perform better than their competitors. Specifically, B-NSGA-II, B-RVEA-i, and B-IBEA obtain better MIGD

values in 13, 9, and 14 of the 15 small-scale instances, respectively. The proportions of the large-scale instances where the three B-DMOEA algorithms, including B-NSGA-II, B-RVEA-i, and B-IBEA, outperform their competitors are 18/24, 16/24, and 14/24 for MIGD, respectively. To intuitively describe the distribution of solutions in three-dimensional objective space, we use D-HRDLB problem S2 as an illustration. Figure 7 plots the final of the six evolutionary algorithms. Figure 7a,b) plot the final solutions of Tr-NSGA-II and B-NSGA-II, respectively. As shown in Figure 7c, we merge the solutions in (a) and (b) to find the nondominated solutions among them. B-NSGA-II can find more nondominated solutions, which means that B-NSGA-II can cover the Pareto front better. From the remaining subplots, we can draw similar conclusions.

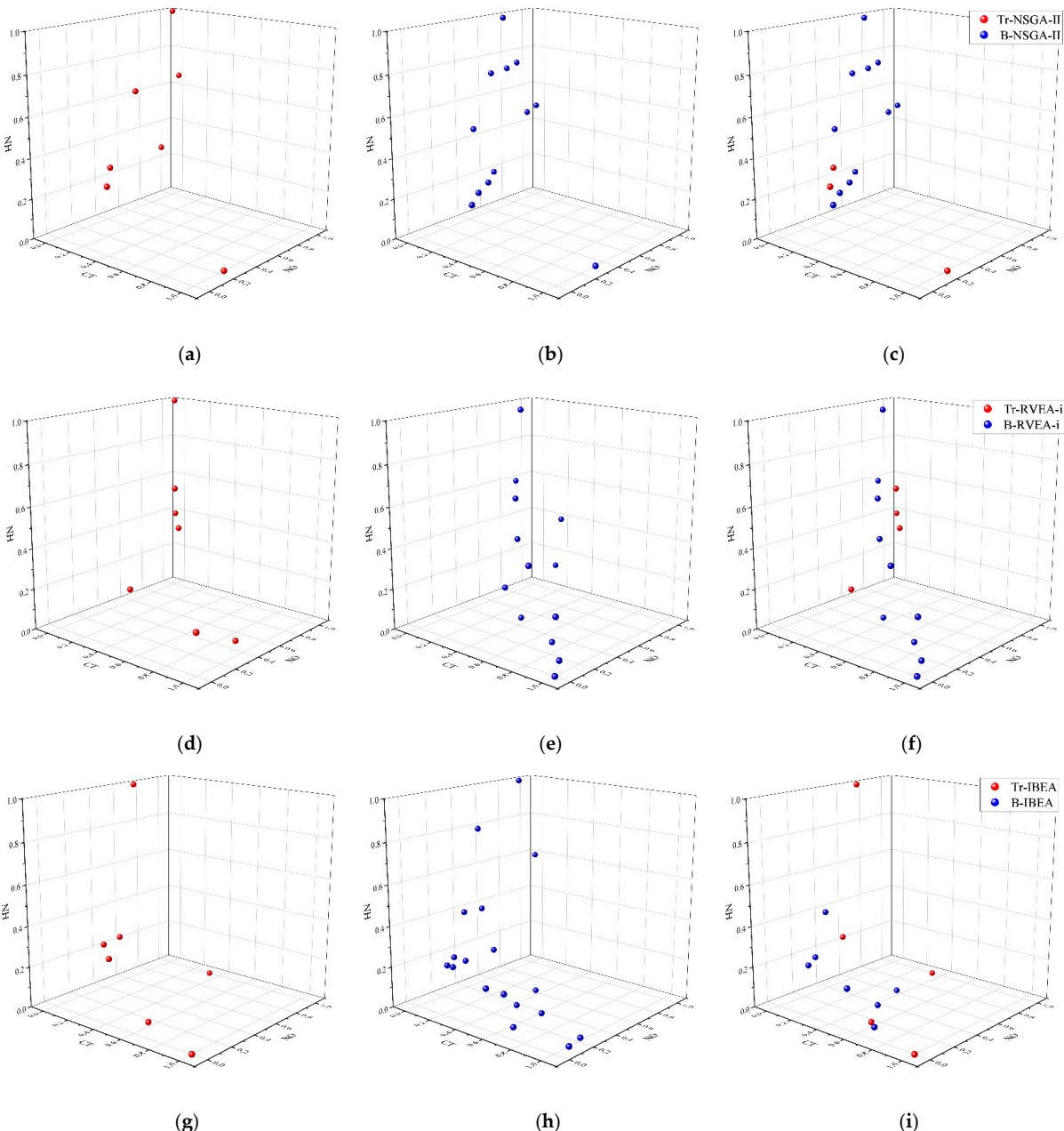

**Figure 7.** Normalised nondominated solution set of problem S2. (**a**) Tr-NSGA-II; (**b**) B-NSGA-II; (**c**) Tr-NSGA-II and B-NSGA-II; (**d**) Tr-RVEA-i; (**e**) B-RVEA-i; (**f**) Tr-RVEA-I and B-RVEA-i; (**g**) Tr-IBEA; (**h**) B-IBEA; (**i**) Tr-IBEA and B-IBEA.

Tables 8 and 9 give the MHV results for all six algorithms on small-scale and large-scale problem instances. Similarly, the three B-DMOEA algorithms outperform their corresponding competitors on most test instances. B-NSGA-II, B-RVEA-i, and B-IBEA achieve better MHV values than their competitors in 13, 13, and 14 of the 15 small-scale instances. Additionally, for large-scale instances, the winning ratio of the three B-DMOEA algorithms B-NSGA-II, B-RVEA-i, and B-IBEA against their competitors is 21 to 3, 18 to 6, and 20 to 4 in the 24 comparisons, respectively.

**Table 8.** Mean and standard deviation values of the MHV metric obtained by comparing algorithms for small-scale instances.

| Problems | Tr-NSGA-II | B-NSGA-II | Tr-RVEA-i | B-RVEA-i | Tr-IBEA | B-IBEA |
|---|---|---|---|---|---|---|
| S1 | $8.8137 \times 10^{-1}$ [$5.2 \times 10^{-2}$] | $\mathbf{1.0172 \times 10^{+00}}$ [$\mathbf{2.0 \times 10^{-2}}$] | $7.6992 \times 10^{-1}$ [$7.4 \times 10^{-2}$] | $\mathbf{7.7664 \times 10^{-1}}$ [$\mathbf{7.3 \times 10^{-2}}$] | $\mathbf{9.1173 \times 10^{-1}}$ [$\mathbf{2.9 \times 10^{-2}}$] | $8.9558 \times 10^{-1}$ [$4.1 \times 10^{-2}$] |
| S2 | $8.7458 \times 10^{-1}$ [$3.5 \times 10^{-2}$] | $\mathbf{9.3420 \times 10^{-1}}$ [$\mathbf{3.0 \times 10^{-2}}$] | $7.5336 \times 10^{-1}$ [$6.7 \times 10^{-2}$] | $\mathbf{7.7190 \times 10^{-1}}$ [$\mathbf{5.4 \times 10^{-2}}$] | $8.8512 \times 10^{-1}$ [$5.4 \times 10^{-2}$] | $\mathbf{1.0468 \times 10^{+00}}$ [$\mathbf{2.5 \times 10^{-2}}$] |
| S3 | $8.6021 \times 10^{-1}$ [$4.6 \times 10^{-2}$] | $\mathbf{8.8963 \times 10^{-1}}$ [$\mathbf{2.4 \times 10^{-2}}$] | $7.7463 \times 10^{-1}$ [$6.0 \times 10^{-2}$] | $\mathbf{7.8602 \times 10^{-1}}$ [$\mathbf{2.7 \times 10^{-2}}$] | $8.7914 \times 10^{-1}$ [$6.8 \times 10^{-2}$] | $\mathbf{9.0052 \times 10^{-1}}$ [$\mathbf{4.2 \times 10^{-2}}$] |
| S4 | $\mathbf{9.0583 \times 10^{-1}}$ [$\mathbf{2.8 \times 10^{-2}}$] | $9.0107 \times 10^{-1}$ [$2.1 \times 10^{-2}$] | $7.8558 \times 10^{-1}$ [$7.1 \times 10^{-2}$] | $\mathbf{7.9593 \times 10^{-1}}$ [$\mathbf{4.9 \times 10^{-2}}$] | $9.0266 \times 10^{-1}$ [$3.6 \times 10^{-2}$] | $\mathbf{9.1661 \times 10^{-1}}$ [$\mathbf{3.5 \times 10^{-2}}$] |
| S5 | $8.6359 \times 10^{-1}$ [$4.0 \times 10^{-2}$] | $\mathbf{8.9617 \times 10^{-1}}$ [$\mathbf{3.8 \times 10^{-2}}$] | $7.8506 \times 10^{-1}$ [$4.9 \times 10^{-2}$] | $\mathbf{8.3523 \times 10^{-1}}$ [$\mathbf{5.1 \times 10^{-2}}$] | $8.6735 \times 10^{-1}$ [$7.7 \times 10^{-2}$] | $\mathbf{8.8879 \times 10^{-1}}$ [$\mathbf{5.5 \times 10^{-2}}$] |
| S6 | $8.9213 \times 10^{-1}$ [$4.8 \times 10^{-2}$] | $\mathbf{9.2073 \times 10^{-1}}$ [$\mathbf{4.4 \times 10^{-2}}$] | $7.9493 \times 10^{-1}$ [$6.8 \times 10^{-2}$] | $\mathbf{8.1352 \times 10^{-1}}$ [$\mathbf{5.9 \times 10^{-2}}$] | $9.5057 \times 10^{-1}$ [$3.9 \times 10^{-2}$] | $\mathbf{1.0468 \times 10^{+00}}$ [$\mathbf{2.5 \times 10^{-2}}$] |
| S7 | $8.4645 \times 10^{-1}$ [$6.3 \times 10^{-2}$] | $\mathbf{8.5654 \times 10^{-1}}$ [$\mathbf{3.3 \times 10^{-2}}$] | $7.7372 \times 10^{-1}$ [$4.3 \times 10^{-2}$] | $\mathbf{7.9620 \times 10^{-1}}$ [$\mathbf{4.9 \times 10^{-2}}$] | $8.5199 \times 10^{-1}$ [$5.6 \times 10^{-2}$] | $\mathbf{8.5667 \times 10^{-1}}$ [$\mathbf{3.4 \times 10^{-2}}$] |
| S8 | $8.9229 \times 10^{-1}$ [$3.4 \times 10^{-2}$] | $\mathbf{8.9895 \times 10^{-1}}$ [$\mathbf{3.3 \times 10^{-2}}$] | $7.7718 \times 10^{-1}$ [$4.6 \times 10^{-2}$] | $\mathbf{7.8784 \times 10^{-1}}$ [$\mathbf{5.4 \times 10^{-2}}$] | $9.1101 \times 10^{-1}$ [$4.5 \times 10^{-2}$] | $\mathbf{9.1815 \times 10^{-1}}$ [$\mathbf{2.7 \times 10^{-2}}$] |
| S9 | $9.0327 \times 10^{-1}$ [$3.8 \times 10^{-2}$] | $\mathbf{9.6682 \times 10^{-1}}$ [$\mathbf{3.3 \times 10^{-2}}$] | $\mathbf{7.8916 \times 10^{-1}}$ [$\mathbf{4.4 \times 10^{-2}}$] | $7.7399 \times 10^{-1}$ [$5.9 \times 10^{-2}$] | $9.3416 \times 10^{-1}$ [$3.1 \times 10^{-2}$] | $\mathbf{9.6207 \times 10^{-1}}$ [$\mathbf{2.1 \times 10^{-2}}$] |
| S10 | $8.9399 \times 10^{-1}$ [$4.7 \times 10^{-2}$] | $\mathbf{9.1839 \times 10^{-1}}$ [$\mathbf{3.8 \times 10^{-2}}$] | $\mathbf{8.1651 \times 10^{-1}}$ [$\mathbf{4.6 \times 10^{-2}}$] | $7.7583 \times 10^{-1}$ [$5.5 \times 10^{-2}$] | $9.2889 \times 10^{-1}$ [$3.3 \times 10^{-2}$] | $\mathbf{9.3775 \times 10^{-1}}$ [$\mathbf{3.2 \times 10^{-2}}$] |
| S11 | $\mathbf{9.1715 \times 10^{-1}}$ [$\mathbf{4.1 \times 10^{-2}}$] | $9.0983 \times 10^{-1}$ [$3.6 \times 10^{-2}$] | $7.8974 \times 10^{-1}$ [$4.0 \times 10^{-2}$] | $\mathbf{8.1380 \times 10^{-1}}$ [$\mathbf{5.4 \times 10^{-2}}$] | $9.3708 \times 10^{-1}$ [$2.6 \times 10^{-2}$] | $\mathbf{9.4471 \times 10^{-1}}$ [$\mathbf{4.5 \times 10^{-2}}$] |
| S12 | $9.0279 \times 10^{-1}$ [$6.2 \times 10^{-2}$] | $\mathbf{9.4832 \times 10^{-1}}$ [$\mathbf{3.9 \times 10^{-2}}$] | $8.0320 \times 10^{-1}$ [$5.4 \times 10^{-2}$] | $\mathbf{8.0803 \times 10^{-1}}$ [$\mathbf{6.5 \times 10^{-2}}$] | $9.2235 \times 10^{-1}$ [$4.1 \times 10^{-2}$] | $\mathbf{9.5697 \times 10^{-1}}$ [$\mathbf{4.6 \times 10^{-2}}$] |
| S13 | $9.0533 \times 10^{-1}$ [$4.4 \times 10^{-2}$] | $\mathbf{9.2646 \times 10^{-1}}$ [$\mathbf{3.8 \times 10^{-2}}$] | $8.0787 \times 10^{-1}$ [$4.5 \times 10^{-2}$] | $\mathbf{8.2723 \times 10^{-1}}$ [$\mathbf{2.9 \times 10^{-2}}$] | $8.7862 \times 10^{-1}$ [$4.8 \times 10^{-2}$] | $\mathbf{9.0766 \times 10^{-1}}$ [$\mathbf{3.8 \times 10^{-2}}$] |
| S14 | $9.0159 \times 10^{-1}$ [$4.3 \times 10^{-2}$] | $\mathbf{9.4480 \times 10^{-1}}$ [$\mathbf{4.3 \times 10^{-2}}$] | $7.8224 \times 10^{-1}$ [$5.2 \times 10^{-2}$] | $\mathbf{7.9015 \times 10^{-1}}$ [$\mathbf{4.7 \times 10^{-2}}$] | $8.7288 \times 10^{-1}$ [$5.2 \times 10^{-2}$] | $\mathbf{9.5582 \times 10^{-1}}$ [$\mathbf{3.3 \times 10^{-2}}$] |
| S15 | $9.0623 \times 10^{-1}$ [$5.2 \times 10^{-2}$] | $\mathbf{9.4391 \times 10^{-1}}$ [$\mathbf{2.6 \times 10^{-2}}$] | $7.8049 \times 10^{-1}$ [$3.9 \times 10^{-2}$] | $\mathbf{8.0042 \times 10^{-1}}$ [$\mathbf{3.8 \times 10^{-2}}$] | $9.4948 \times 10^{-1}$ [$4.4 \times 10^{-2}$] | $\mathbf{9.6361 \times 10^{-1}}$ [$\mathbf{4.3 \times 10^{-2}}$] |

The better mean for each instance is highlighted in boldface.

**Table 9.** Mean and standard deviation values of the MHV metric obtained by comparing algorithms for large-scale instances.

| Problems | Tr-NSGA-II | B-NSGA-II | Tr-RVEA-i | B-RVEA-i | Tr-IBEA | B-IBEA |
|---|---|---|---|---|---|---|
| L1 | $9.0036 \times 10^{-1}$ [$3.5 \times 10^{-2}$] | $\mathbf{9.1018 \times 10^{-1}}$ [$\mathbf{2.2 \times 10^{-2}}$] | $7.6746 \times 10^{-1}$ [$6.0 \times 10^{-2}$] | $\mathbf{8.2244 \times 10^{-1}}$ [$\mathbf{6.9 \times 10^{-2}}$] | $8.8050 \times 10^{-1}$ [$4.5 \times 10^{-2}$] | $\mathbf{9.0750 \times 10^{-1}}$ [$\mathbf{3.7 \times 10^{-2}}$] |
| L2 | $\mathbf{9.0271 \times 10^{-1}}$ [$\mathbf{2.7 \times 10^{-2}}$] | $8.9514 \times 10^{-1}$ [$5.4 \times 10^{-2}$] | $7.8644 \times 10^{-1}$ [$6.2 \times 10^{-2}$] | $\mathbf{8.0711 \times 10^{-1}}$ [$\mathbf{5.6 \times 10^{-2}}$] | $8.8747 \times 10^{-1}$ [$7.9 \times 10^{-2}$] | $\mathbf{9.0314 \times 10^{-1}}$ [$\mathbf{5.0 \times 10^{-2}}$] |
| L3 | $\mathbf{9.1960 \times 10^{-1}}$ [$\mathbf{3.4 \times 10^{-2}}$] | $8.7439 \times 10^{-1}$ [$4.2 \times 10^{-2}$] | $8.1990 \times 10^{-1}$ [$5.0 \times 10^{-2}$] | $8.0815 \times 10^{-1}$ [$4.8 \times 10^{-2}$] | $\mathbf{9.2754 \times 10^{-1}}$ [$\mathbf{4.9 \times 10^{-2}}$] | $8.9086 \times 10^{-1}$ [$5.8 \times 10^{-2}$] |
| L4 | $9.0131 \times 10^{-1}$ [$5.1 \times 10^{-2}$] | $\mathbf{9.0945 \times 10^{-1}}$ [$\mathbf{8.0 \times 10^{-2}}$] | $\mathbf{8.2961 \times 10^{-1}}$ [$\mathbf{2.5 \times 10^{-2}}$] | $8.0492 \times 10^{-1}$ [$6.5 \times 10^{-2}$] | $8.9199 \times 10^{-1}$ [$5.6 \times 10^{-2}$] | $\mathbf{9.8785 \times 10^{-1}}$ [$\mathbf{5.4 \times 10^{-2}}$] |
| L5 | $8.9207 \times 10^{-1}$ [$6.4 \times 10^{-2}$] | $\mathbf{9.0436 \times 10^{-1}}$ [$\mathbf{5.1 \times 10^{-2}}$] | $8.0022 \times 10^{-1}$ [$6.5 \times 10^{-2}$] | $\mathbf{8.0652 \times 10^{-1}}$ [$\mathbf{4.9 \times 10^{-2}}$] | $9.0361 \times 10^{-1}$ [$5.4 \times 10^{-2}$] | $\mathbf{9.0530 \times 10^{-1}}$ [$\mathbf{3.5 \times 10^{-2}}$] |
| L6 | $8.8144 \times 10^{-1}$ [$6.5 \times 10^{-2}$] | $\mathbf{9.0795 \times 10^{-1}}$ [$\mathbf{2.6 \times 10^{-2}}$] | $7.9694 \times 10^{-1}$ [$6.4 \times 10^{-2}$] | $\mathbf{8.2378 \times 10^{-1}}$ [$\mathbf{4.0 \times 10^{-2}}$] | $8.9411 \times 10^{-1}$ [$5.1 \times 10^{-2}$] | $\mathbf{9.2053 \times 10^{-1}}$ [$\mathbf{2.5 \times 10^{-2}}$] |
| L7 | $8.8145 \times 10^{-1}$ [$5.8 \times 10^{-2}$] | $\mathbf{8.9781 \times 10^{-1}}$ [$\mathbf{3.4 \times 10^{-2}}$] | $7.8760 \times 10^{-1}$ [$1.1 \times 10^{-1}$] | $\mathbf{8.2209 \times 10^{-1}}$ [$\mathbf{7.8 \times 10^{-2}}$] | $8.9717 \times 10^{-1}$ [$4.4 \times 10^{-2}$] | $\mathbf{9.2491 \times 10^{-1}}$ [$\mathbf{3.0 \times 10^{-2}}$] |
| L8 | $8.5915 \times 10^{-1}$ [$4.1 \times 10^{-2}$] | $\mathbf{8.8811 \times 10^{-1}}$ [$\mathbf{4.0 \times 10^{-2}}$] | $\mathbf{8.2064 \times 10^{-1}}$ [$\mathbf{5.1 \times 10^{-2}}$] | $8.0646 \times 10^{-1}$ [$5.9 \times 10^{-2}$] | $8.7540 \times 10^{-1}$ [$5.0 \times 10^{-2}$] | $\mathbf{8.9641 \times 10^{-1}}$ [$\mathbf{4.1 \times 10^{-2}}$] |
| L9 | $8.7632 \times 10^{-1}$ [$4.4 \times 10^{-2}$] | $\mathbf{9.0017 \times 10^{-1}}$ [$\mathbf{1.9 \times 10^{-2}}$] | $7.9540 \times 10^{-1}$ [$7.8 \times 10^{-2}$] | $\mathbf{8.2099 \times 10^{-1}}$ [$\mathbf{6.6 \times 10^{-2}}$] | $8.5812 \times 10^{-1}$ [$7.8 \times 10^{-2}$] | $\mathbf{8.8288 \times 10^{-1}}$ [$\mathbf{7.0 \times 10^{-2}}$] |
| L10 | $8.6768 \times 10^{-1}$ [$5.8 \times 10^{-2}$] | $\mathbf{8.7925 \times 10^{-1}}$ [$\mathbf{3.6 \times 10^{-2}}$] | $8.1317 \times 10^{-1}$ [$5.4 \times 10^{-2}$] | $\mathbf{8.2274 \times 10^{-1}}$ [$\mathbf{4.6 \times 10^{-2}}$] | $9.0848 \times 10^{-1}$ [$5.8 \times 10^{-2}$] | $\mathbf{9.0875 \times 10^{-1}}$ [$\mathbf{3.6 \times 10^{-2}}$] |
| L11 | $\mathbf{8.9713 \times 10^{-1}}$ [$\mathbf{2.5 \times 10^{-2}}$] | $8.8994 \times 10^{-1}$ [$4.9 \times 10^{-2}$] | $\mathbf{8.2921 \times 10^{-1}}$ [$\mathbf{3.9 \times 10^{-2}}$] | $8.1423 \times 10^{-1}$ [$4.3 \times 10^{-2}$] | $\mathbf{9.2999 \times 10^{-1}}$ [$\mathbf{5.5 \times 10^{-2}}$] | $9.0812 \times 10^{-1}$ [$6.5 \times 10^{-2}$] |
| L12 | $9.0475 \times 10^{-1}$ [$4.7 \times 10^{-2}$] | $\mathbf{9.0880 \times 10^{-1}}$ [$\mathbf{2.4 \times 10^{-2}}$] | $8.1330 \times 10^{-1}$ [$7.4 \times 10^{-2}$] | $\mathbf{8.4537 \times 10^{-1}}$ [$\mathbf{3.3 \times 10^{-2}}$] | $8.9468 \times 10^{-1}$ [$4.2 \times 10^{-2}$] | $\mathbf{9.2687 \times 10^{-1}}$ [$\mathbf{3.9 \times 10^{-2}}$] |
| L13 | $9.0550 \times 10^{-1}$ [$5.3 \times 10^{-2}$] | $\mathbf{9.2308 \times 10^{-1}}$ [$\mathbf{3.0 \times 10^{-2}}$] | $8.2124 \times 10^{-1}$ [$7.1 \times 10^{-2}$] | $\mathbf{8.2749 \times 10^{-1}}$ [$\mathbf{4.4 \times 10^{-2}}$] | $9.0963 \times 10^{-1}$ [$7.0 \times 10^{-2}$] | $\mathbf{9.3389 \times 10^{-1}}$ [$\mathbf{3.5 \times 10^{-2}}$] |
| L14 | $8.9452 \times 10^{-1}$ [$3.1 \times 10^{-2}$] | $\mathbf{9.0445 \times 10^{-1}}$ [$\mathbf{2.4 \times 10^{-2}}$] | $8.0014 \times 10^{-1}$ [$5.4 \times 10^{-2}$] | $\mathbf{8.1013 \times 10^{-1}}$ [$\mathbf{5.8 \times 10^{-2}}$] | $9.0709 \times 10^{-1}$ [$4.3 \times 10^{-2}$] | $\mathbf{9.4785 \times 10^{-1}}$ [$\mathbf{3.7 \times 10^{-2}}$] |
| L15 | $8.9159 \times 10^{-1}$ [$5.9 \times 10^{-2}$] | $\mathbf{9.1480 \times 10^{-1}}$ [$\mathbf{3.9 \times 10^{-2}}$] | $7.8091 \times 10^{-1}$ [$5.0 \times 10^{-2}$] | $\mathbf{8.1870 \times 10^{-1}}$ [$\mathbf{6.2 \times 10^{-2}}$] | $9.1601 \times 10^{-1}$ [$4.1 \times 10^{-2}$] | $\mathbf{9.3450 \times 10^{-1}}$ [$\mathbf{3.2 \times 10^{-2}}$] |
| L16 | $8.8373 \times 10^{-1}$ [$5.3 \times 10^{-2}$] | $\mathbf{9.2835 \times 10^{-1}}$ [$\mathbf{4.4 \times 10^{-2}}$] | $8.1387 \times 10^{-1}$ [$6.5 \times 10^{-2}$] | $\mathbf{9.0778 \times 10^{-1}}$ [$\mathbf{4.8 \times 10^{-2}}$] | $9.0778 \times 10^{-1}$ [$4.8 \times 10^{-2}$] | $\mathbf{8.8873 \times 10^{-1}}$ [$\mathbf{5.9 \times 10^{-2}}$] |
| L17 | $9.0005 \times 10^{-1}$ [$4.4 \times 10^{-2}$] | $\mathbf{9.0100 \times 10^{-1}}$ [$\mathbf{4.1 \times 10^{-2}}$] | $8.0976 \times 10^{-1}$ [$4.4 \times 10^{-2}$] | $\mathbf{8.1575 \times 10^{-1}}$ [$\mathbf{4.1 \times 10^{-2}}$] | $9.1786 \times 10^{-1}$ [$3.8 \times 10^{-2}$] | $\mathbf{9.2020 \times 10^{-1}}$ [$\mathbf{4.0 \times 10^{-2}}$] |
| L18 | $9.0482 \times 10^{-1}$ [$4.7 \times 10^{-2}$] | $\mathbf{9.3028 \times 10^{-1}}$ [$\mathbf{3.0 \times 10^{-2}}$] | $8.2526 \times 10^{-1}$ [$3.6 \times 10^{-2}$] | $\mathbf{8.4592 \times 10^{-1}}$ [$\mathbf{5.4 \times 10^{-2}}$] | $9.3418 \times 10^{-1}$ [$4.2 \times 10^{-2}$] | $\mathbf{9.5725 \times 10^{-1}}$ [$\mathbf{3.5 \times 10^{-2}}$] |
| L19 | $9.1043 \times 10^{-1}$ [$5.5 \times 10^{-2}$] | $\mathbf{9.1367 \times 10^{-1}}$ [$\mathbf{4.0 \times 10^{-2}}$] | $8.0992 \times 10^{-1}$ [$4.9 \times 10^{-2}$] | $\mathbf{8.2194 \times 10^{-1}}$ [$\mathbf{4.8 \times 10^{-2}}$] | $\mathbf{9.4361 \times 10^{-1}}$ [$\mathbf{3.5 \times 10^{-2}}$] | $8.9129 \times 10^{-1}$ [$5.7 \times 10^{-2}$] |
| L20 | $8.8647 \times 10^{-1}$ [$4.4 \times 10^{-2}$] | $\mathbf{9.0433 \times 10^{-1}}$ [$\mathbf{4.6 \times 10^{-2}}$] | $\mathbf{8.2320 \times 10^{-1}}$ [$\mathbf{4.2 \times 10^{-2}}$] | $8.0300 \times 10^{-1}$ [$5.7 \times 10^{-2}$] | $8.9220 \times 10^{-1}$ [$8.4 \times 10^{-2}$] | $\mathbf{9.1475 \times 10^{-1}}$ [$\mathbf{6.4 \times 10^{-2}}$] |
| L21 | $8.5629 \times 10^{-1}$ [$6.7 \times 10^{-2}$] | $\mathbf{8.8725 \times 10^{-1}}$ [$\mathbf{5.1 \times 10^{-2}}$] | $8.1573 \times 10^{-1}$ [$8.2 \times 10^{-2}$] | $\mathbf{8.2549 \times 10^{-1}}$ [$\mathbf{4.9 \times 10^{-2}}$] | $9.2198 \times 10^{-1}$ [$3.6 \times 10^{-2}$] | $\mathbf{9.2877 \times 10^{-1}}$ [$\mathbf{3.8 \times 10^{-2}}$] |
| L22 | $8.0581 \times 10^{-1}$ [$6.0 \times 10^{-2}$] | $\mathbf{8.4296 \times 10^{-1}}$ [$\mathbf{5.3 \times 10^{-2}}$] | $7.7601 \times 10^{-1}$ [$4.6 \times 10^{-2}$] | $\mathbf{8.3737 \times 10^{-1}}$ [$\mathbf{6.5 \times 10^{-2}}$] | $9.1162 \times 10^{-1}$ [$5.7 \times 10^{-2}$] | $\mathbf{9.3818 \times 10^{-1}}$ [$\mathbf{6.3 \times 10^{-2}}$] |
| L23 | $8.4144 \times 10^{-1}$ [$4.0 \times 10^{-2}$] | $\mathbf{8.9131 \times 10^{-1}}$ [$\mathbf{3.9 \times 10^{-2}}$] | $7.8913 \times 10^{-1}$ [$4.4 \times 10^{-2}$] | $\mathbf{8.1356 \times 10^{-1}}$ [$\mathbf{3.6 \times 10^{-2}}$] | $9.1316 \times 10^{-1}$ [$5.8 \times 10^{-2}$] | $\mathbf{9.2955 \times 10^{-1}}$ [$\mathbf{4.5 \times 10^{-2}}$] |
| L24 | $8.6985 \times 10^{-1}$ [$5.5 \times 10^{-2}$] | $\mathbf{8.8090 \times 10^{-1}}$ [$\mathbf{5.0 \times 10^{-2}}$] | $\mathbf{8.0074 \times 10^{-1}}$ [$\mathbf{5.2 \times 10^{-2}}$] | $7.3800 \times 10^{-1}$ [$5.3 \times 10^{-2}$] | $8.9682 \times 10^{-1}$ [$5.5 \times 10^{-2}$] | $\mathbf{8.9945 \times 10^{-1}}$ [$\mathbf{4.2 \times 10^{-2}}$] |

The better mean for each instance is highlighted in boldface.

These experimental results demonstrate that B-DMOEA can perform better than Tr-DMOEA. B-DMOEA combines the marginal and conditional distributions to learn the common features between the domains, making knowledge transfer more effective and further improving the performance of MOEAs compared with Tr-DMOEA. In addition, whether in small-scale or large-scale problem instances, the three B-DMOEA algorithms can win most of the comparisons. However, the results also show that the advantages of the three B-DMOEA algorithms are more evident in small-scale instances. One important reason for this occurrence is that in some large-scale instances, due to the relatively significant difference in product quality, the distribution differences between domains are more manifested in the marginal distributions. In contrast to the small-scale instances, the conditional distributions should be given more attention due to the minor differences in product quality. In this case, B-DMEOA, considering the conditional distribution, performs better.

### 7. Conclusions

This paper studied a new dynamic human-robot collaborative disassembly line balancing problem, D-HRDLB, in which multiple humans and robots can perform multiple disassembly operations collaboratively in any workstation of the disassembly line. Various dynamic factors in the disassembly environment, especially uncertain product quality, can be responded to promptly. Therefore, D-HRDLB is closer to the actual disassembly environment in remanufacturing enterprises than the existing DLBPs. Since the existing disassembly process models cannot describe the human-robot collaborative mode and the time-varying environment, a new task-based dynamic disassembly process model, TD-TAOG, was proposed. Then, a formal definition of the model was illustrated.

Compared with existing static or quasi-static disassembly process models such as TAOG, the unique features of TD-TAOG are manifested in two aspects: (i) in the selection of basic elements, TAOG chooses the disassembly operation, while TD-TAOG chooses a more fine-grained disassembly task; and (ii) TAOG can only be used to describe products of definite quality, whereas TD-TAOG can characterise the time-varying properties of tasks affected by uncertain product quality. Furthermore, based on TD-TAOG, we formalised the decision variables, the objective functions, and the constraints in D-HRDLB, and we established a multiobjective optimisation model for D-HRDLB. Compared with existing nondeterministic optimisation models for DLBPs, this model is a dynamic optimisation model that can better resolve various uncertain factors. Since D-HRDLB has been identified as NP-hard, a tracking multiobjective dynamic evolutionary algorithm B-DMOEA based on evolutionary dynamic optimisation and feature-based transfer learning was proposed to track and respond to dynamic factors. We analysed the algorithm framework, the initial population generation assisted by transfer learning, the solution encoding and decoding, and the solution variation. In B-DMOEA, the initial population generation is inspired by Tr-DMOEA. However, B-DMOEA differs from Tr-DMOEA in three aspects: (i) B-DMOEA adopts a presearch strategy based on MOEAs, which can improve the authenticity of the sample distribution; (ii) unlike Tr-DMOEA, which selects the previous historical environment as the source domain, B-DMOEA selects the environment that is most similar to the current environment in the historical environment as the source domain of knowledge transfer; and (iii) B-DMOEA considers the marginal distribution and the conditional distribution of the sample sets, and it can adjust the weights of these two distributions according to the distribution difference between the domains. Tr-DMOEA only considers the marginal distribution of the sample set. These properties of B-DMOEA can effectively reduce the probability of negative transfer.

To investigate the behaviour of B-DMOEA on D-HRDLB, a set of D-HRDLB problem instances containing small-scale and large-scale instances were generated as the benchmark. Fifteen small-scale instances showed relatively small environmental changes in the set, and 24 large-scale instances displayed relatively significant environmental changes. Based on the generated instance set, we selected three representative MOEAs, including NSGA-II, RVEA-i, and IBEA, incorporated them into Tr-DMOEA and B-DMOEA, respectively, and compared the performance of these obtained algorithms through experiments. Experimental results show, first, that in B-DMOEA, the value $\mu$ has a significant impact on the quality of the initial population. Regarding most small-scale instances, because the environment has not changed much, the influence of the conditional distribution on the effect of transfer learning is more significant. In contrast, considering most large-scale instances, due to significant environmental changes, the marginal distribution has a more significant impact on the effect of transfer learning. Second, B-DMOEA can perform better than Tr-DMOEA.

The various environmental uncertainties in remanufacturing enterprises are obstacles to the practical application related to DLBPs. One area for further investigation is to develop dynamic disassembly line balancing optimisation models that characterise more dynamic factors. In addition, more transfer learning methods, such as sample transfer, model transfer, and relationship transfer, can be combined with evolutionary dynamic optimisation to develop more effective algorithms for D-HRDLB.

**Author Contributions:** L.J.: conceptualisation, formal analysis, investigation, methodology, and writing—original draft. X.Z.: formal analysis, resources, supervision, and writing—original draft. Y.F.: formal analysis, investigation, methodology, and writing—original draft. D.T.P.: formal analysis, methodology, resources, visualisation, and writing—review & editing. All authors have read and agreed to the published version of the manuscript.

**Funding:** This research was funded by the National Natural Science Foundation of China under Grant No. 52075402.

**Institutional Review Board Statement:** Not applicable.

**Informed Consent Statement:** Not applicable.

**Data Availability Statement:** The datasets obtained during the current work are available from the corresponding author upon request.

**Acknowledgments:** The authors wish to thank the editor and the anonymous referees for their comments, which have helped to improve this paper.

**Conflicts of Interest:** The datasets obtained during the current work are available from the corresponding author upon request.

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
