# Peer review of "Transfer Learning-Assisted Evolutionary Dynamic Optimisation for Dynamic Human-Robot Collaborative Disassembly Line Balancing"

_applsci, doi:10.3390/app122111008_

Round 1

Reviewer 1 Report

The present work is current and well presented.The work should be improved, particularly in chapter 4, or rather I would make it a little more concise in the exposition and therefore easier to read and understand (I have lost the thread of the discussion several times). Ultimately the job is well done and worthy of publication. In addition, it would be better to improve your English a little.

I would add some quotes in particular on the structure of the individual robots:

Cammarata, A., Lacagnina, M., Sinatra, R. "Closed-form solutions for the inverse kinematics of the Agile Eye with constraint errors on the revolute joint axes (2016)", IEEE International Conference on Intelligent Robots and Systems, 2016-November, art. no. 7759073, pp. 317-322. 

Reviewer 2 Report

The article deals with an interesting and current subject related to collaborative disassembly line balancing. The article provides extensive overview of the state of the art in the area of disassembly process modeling and disassembly line balancing under uncertainty. However, this review does not apply everywhere exactly to all literature items, as some items are cited collectively without analyzing them (e.g. p. 4, line 188).

A task-based dynamic disassembly process model is proposed and discussed on the example of flashlight disassembly.

The problem of dynamic human-robot collaborative disassembly line balancing was analyzed, assuming the presence of 3 workstations on the line.

The proposal of transfer learning-assisted dynamic evolutionary algorithm solving the problem of disassembly line balancing was presented and component algorithms were discussed. The solution of the analyzed problem of disassembly line balancing and the results of computational experiments were presented, including the issue of the sensitivity of the model parameters. The efficiency of the obtained solutions was also assessed.

Finally, selected conclusions resulting from the conducted research are presented and potential directions for further work are indicated.

Recommendations for improving the article:

1.      Authors should discuss in more detail the selected references that are currently cited collectively.

2.      Authors should correct the description of their method. It should be more understandable to readers so that they can also repeat the research presented in the paper. In particular, an introduction should be added to section 5. Transfer learning-assisted dynamic evolutionary algorithm, where the algorithm is presented at the beginning without proper introduction.

3.      The results in the form of Table 7, and especially 8-11, are hardly readable for humans and difficult to interpret. Authors should consider moving them to the appendix. Authors should consider presenting these results in form of graphs, if possible.

Reviewer 3 Report

The paper is written well, and the technical content is also acceptable. The following are my comments. 

1. The abstract looks vague. Short the abstract and include only important statements related to novelty and findings.

2. The authors have done a detailed literature study. However, the literature study on the multiobjective algorithm is not there, and it is not fair too. For instance, you refer to 10.1016/j.engappai.2021.104480, 10.1038/s41598-021-99617-x, 10.1016/j.knosys.2021.106856, 10.3389/fmars.2019.00017, 10.1016/j.eswa.2020.114150, etc.

3.  It is suggested to deposit the code in any one repository so that it will be helpful for the reviewers to test the results presented.

4. Why have the authors not presented the Pareto front curve? It is necessary to analyze how well the solutions are distributed. 

5. The authors have used only IGD and HV metrics to analyze the performance. Why not other metrics, such as Spread, Spacing, GD, etc. There are more than 50 metrics are available.

6. The authors have calculated IGD; however, they didn't discuss anything about the true Pareto front. Have you generated a true Pareto? If so, how have you generated it? 

7. It is good that you have selected all the good algorithms for performance comparison. 

8. Correct all the grammatical mistakes and typo errors

9. Apart from these, there are no technical defects in the application selected and the way it has been presented. 

Reviewer 4 Report

I am specialized in balancing and modelling problem. This paper is very well structured with a large workload, a separate literature review section and description of the research methodology, a discussion section and a well-developed conclusion. The abstract also mentions the research background, research methodology, research objectives and conclusions in a reasonable way.

This paper was very comprehensive in studying the human-robot collaborative disassembly line balancing problem and I enjoyed it.

However, I still have some questions and suggestions after review the manuscript:

1. I suggested the author give the managerial insights for the real-world disassembly line balancing.

2. The reasons for algorithm parameters setting should be discussed.

Additional comments

1.      In the Introduction, the author(s) should state more clearly the importance of the research.

2.      A clearer research gap should be summarized.

3.      Could you please provide more literatures about the human-robot collaborative disassembly or assembly line balancing problem.

4.      Please give a logic flow chart for the study.

5.      A pseudocode is a nice-to-have the illustrate the scheduling algorithm.

6.      Please make sure you have clarified your contribution to knowledge and the novelty of the research. In addition, compare your research against the identified literature and competition in the field.

Round 2

Reviewer 2 Report

The revised version of the manuscript takes into account most of the previous remarks.
Therefore, it can be published in the current version.

Reviewer 3 Report

The authors have taken an effort and responded to all my comments. I am satisfied with the responses and I hope the paper is improved from its original version. I recommend the paper for further proceedings.